# Perinatal thymic-derived CD8αβ-expressing γδ T cells are innate IFN-γ producers that expand in IL-7R–STAT5B-driven neoplasms

Nital Sumaria[1,11], Gina J. Fiala [2,3,4,11] ✉, Daniel Inácio[2], Marta Curado-Avelar[2], Ana Cachucho[2], Rúben Pinheiro[2], Robert Wiesheu [5,6], Shunsuke Kimura [7], Lucien Courtois[8], Birte Blankenhaus[2], Julie Darrigues[2], Tobias Suske[9], Afonso R. M. Almeida[2], Susana Minguet [3,4,10], Vahid Asnafi [8], Ludovic Lhermitte[8], Charles G. Mulligan [7], Seth B. Coffelt [5,6], Richard Moriggl [9], João T. Barata[2,11], Daniel J. Pennington [1,11] ✉ & Bruno Silva-Santos[2,11]

The contribution of γδ T cells to immune responses is associated with rapid secretion of interferon-γ (IFN-γ). Here, we show a perinatal thymic wave of innate IFN-γ-producing γδ T cells that express CD8αβ heterodimers and expand in preclinical models of infection and cancer. Optimal CD8αβ⁺ γδ T cell development is directed by low T cell receptor signaling and through provision of interleukin (IL)-4 and IL-7. This population is pathologically relevant as overactive, or constitutive, IL-7R–STAT5B signaling promotes a supraphysiological accumulation of CD8αβ⁺ γδ T cells in the thymus and peripheral lymphoid organs in two mouse models of T cell neoplasia. Likewise, CD8αβ⁺ γδ T cells define a distinct subset of human T cell acute lymphoblastic leukemia pediatric patients. This work characterizes the normal and malignant development of CD8αβ⁺ γδ T cells that are enriched in early life and contribute to innate IFN-γ responses to infection and cancer.

γδ T cells are prototypic unconventional lymphocytes that display myriad nonredundant functions in tissue homeostasis, and in immune responses against pathogens or tumors[1]. γδ T cells provide a critical early source of cytokines, notably IFN-γ and IL-17A, that have direct effects in tissues and on downstream adaptive immune responses. In particular, IFN-γ-producing γδ (γδ[IFN]) T cells have generated much recent interest due to their potent antitumor functions in mice and humans, which have encouraged the development of γδ T cell-based therapies in various cancer settings[2].

Despite this exciting potential, much about the γδ[IFN] T cell compartment remains unclear, particularly in preclinical models where their IL-17-producing (γδ[17]) counterparts have gathered disproportionate attention[1]. Thus, while various γδ T cell subsets have been reported to have IFN-γ-secreting potential, including dendritic epidermal T cells (DETCs)[3], thymus-leukemia antigen (TL)-specific γδ T cells[4], natural killer T (NKT)-like γδ T cells[5], thymic ligand-independent γδ[IFN] cells[6] and CD8αα⁺ intraepithelial lymphocytes[7], the extent to which these subsets contribute to IFN-γ-dependent immune responses remains

[1]Blizard Institute, Barts and The London School of Medicine, Queen Mary University of London, London, UK. [2]Instituto de Medicina Molecular João Lobo Antunes, Faculdade de Medicina, Universidade de Lisboa, Lisboa, Portugal. [3]Faculty of Biology, University of Freiburg, Freiburg, Germany. [4]Signalling Research Centres BIOSS and CIBSS, University of Freiburg, Freiburg, Germany. [5]Cancer Research UK Scotland Institute, Glasgow, UK. [6]School of Cancer Sciences, University of Glasgow, Glasgow, UK. [7]St. Jude's Children's Research Hospital, Memphis, TN, USA. [8]Hôpital Necker Enfants-Malades, Université de Paris, Paris, France. [9]Department of Biosciences and Medical Biology, Paris Lodron University of Salzburg, Salzburg, Austria. [10]Center of Chronic Immunodeficiency CCI, University Clinics and Medical Faculty, Freiburg, Germany. [11]These authors contributed equally: Nital Sumaria, Gina J. Fiala, João T. Barata, Daniel J. Pennington, Bruno Silva-Santos. ✉e-mail: gina.fiala@biologie.uni-freiburg.de; d.pennington@qmul.ac.uk

to be determined. This may be particularly relevant at the neonatal stage, when γδ T cells play key roles before the definitive development of αβ T cells[8]. Indeed, whereas γδ[17] cells have been consistently implicated in this period[1], an equivalent γδ[IFN] T cell subset is yet to be characterized.

An experimental window into this peripheral γδ T cell subset complexity is the study of the perinatal thymus, where the majority of γδ T cells develop and, importantly, where they commit to subsequent cytokine-secreting effector function (unlike their αβ T cell counterparts)[9]. To utilize this perinatal window, we previously proposed a developmental framework to describe the stepwise thymic generation of distinct γδ T cell subsets that permits mechanistic interrogation of the processes driving acquisition of γδ T cell effector characteristics and subsequent functional capacities[10]. For example, strong γδ T cell receptor (TCR) signaling was shown to drive γδ progenitors into the CD45RB-expressing IFN-γ-pathway but was not compatible with development of IL-17-producing γδ T cells[10–12], enforcing developmental fates that are underpinned by distinct metabolic programming[13].

The continued utility of this approach also recently identified a thymic subset of CD8β[+]Ly6a[+] γδ T cells that were notably expanded in thymic organ culture when TCRγδ-induced PI3K signaling was reduced[12]. This evoked the longstanding observation of γδ T cells expressing CD8αβ heterodimers in the fetal thymus, which expanded upon provision of IL-7 (ref. 14). Interestingly, human γδ T cells expressing CD8αβ heterodimers were independently reported in various and severe pathological settings. In a study on inflammatory bowel disease, CD8αβ[+] γδ T cells were shown to display a cytotoxic type 1 effector profile, including the production of IFN-γ, were inversely correlated with the degree of disease activity and were restored to normal levels upon anti-tumor necrosis factor (TNF) therapy[15]. In a more recent report, CD8αβ[+] γδ T cells were identified as a fraction of a distinctive CD8[+] γδ T cell subset with 'NK-like' cytotoxic type 1 features that expanded in chronic *Mycobacterium tuberculosis* infection, as well as in other chronic (but not acute) inflammatory conditions, such as cardiovascular disease and cancer[16].

While the existence of CD8αβ[+] γδ T cells is now undisputed, their development and functional properties are not well characterized. Here, we demonstrate that CD8αβ[+] γδ T cells develop through a unique thymic pathway and populate multiple peripheral tissues in neonatal mice, persisting into adulthood, and expanding upon tumor challenge or infection. Moreover, they possess prominent innate-like characteristics that include expression of the transcription factor eomesodermin (Eomes), and robust TCR-independent secretion of IFN-γ in response to the innate cytokines IL-12 and IL-18. Efficient generation of this polyclonal subset along a distinct developmental trajectory requires a lower TCRγδ signal strength in collaboration with IL-4, that upregulates Eomes. Notably, IL-7 is the most potent driver of CD8αβ[+] γδ T cell expansion, which can become pathological upon aberrant IL-7R signaling, driven by the transcription factor STAT5B, as demonstrated in two preclinical models of T cell malignancy. Finally, CD8αβ[+] γδ T cells are conspicuously found in a subset of pediatric patients with T cell acute lymphoblastic leukemia (T-ALL), thus attesting their relevance in a human cancer context.

## Results
### CD8αβ[+] γδ T cells respond to tumor challenge and malaria infection
Following recent reports on CD8αβ[+] γδ T cells in human pathology[15,16], we ascertained their presence and functional contribution in preclinical mouse models of disease, aiming to further dissect the biology of this new γδ T cell subset. We chose cancer and malaria models based on the well-established roles of γδ T cells in these diseases[13,17]. A stringent gating strategy for γδ T cells avoided any artifacts due to the abundance of αβ T cells when detecting subsets of γδ T cells based on CD8α versus CD8β expression by flow cytometry analysis (Extended Data

Fig. 1a). First, we analyzed a breast cancer model based on the orthotopic injection of E0771 cells[13] (Fig. 1a). After gating out IL-17-producing CD44[+]CD45RB[−] (γδ[17]) T cells, we clearly detected CD8αβ[+] γδ T cells as tumor-infiltrating lymphocytes (TILs), albeit with notably less abundance than CD8[−] γδ T cells (Fig. 1b and Extended Data Fig. 1b). However, CD8αβ[+] γδ T cells accounted for almost half the IFN-γ-producing γδ TILs, similar to the CD8[−] subset, and in stark contrast with CD8αα[+] γδ T cells (Fig. 1c). This major contribution was selective to the tumor bed and not found in distal lymph nodes (LNs; Extended Data Fig. 1c), suggesting a preferential expansion of IFN-γ-expressing CD8αβ[+] γδ T cells within E0771 tumors. We then performed a similar analysis of various organs (liver, spleen and peripheral LNs) of C57BL/6 (B6) mice at day 5 after infection with *Plasmodium berghei* ANKA sporozoites (Fig. 1d) and, similarly, detected three subsets of γδ T cells based on CD8α versus CD8β expression (Extended Data Fig. 1d). Compared to noninfected controls, infected animals showed expansions of all three γδ T cell subsets (Fig. 1e), especially in the spleen where cell proliferation (as assessed by Ki-67 staining) was increased. As expected[17], the functional response was dominated by IFN-γ (rather than IL-17) production by all γδ T cell subsets, but particularly enriched in CD8αβ[+] γδ T cells (Fig. 1f). These data demonstrate that mouse CD8αβ[+] γδ T cells respond—that is, expand and produce IFN-γ—to both cancer and infectious challenges.

### CD8αβ[+] γδ T cells are a stable subset of innate IFN-γ producers
Having found CD8αβ[+] γδ T cells in selected tissues of the previous disease models, we next assessed the repertoire of organs they naturally populate at steady state and at different stages of mouse ontogeny. We found discrete CD8αβ[+] γδ populations in every organ analyzed, with a selective enrichment relative to other γδ T cell subsets in LNs, spleen and lungs of adult wild-type (WT) mice (Fig. 2a). Interestingly, CD8αβ[+] γδ populations made up a sizable proportion (~24% in spleen, ~13% in lungs and ~35% in small intestine), or the majority (~52% in LN), of γδ T cells in neonatal mice (Fig. 2a).

To assess the functional contributions of the different peripheral γδ T cell subsets at steady state, we performed intracellular staining for IFN-γ and IL-17A in neonatal γδ T cells upon stimulation in vitro with phorbol myristate acetate (PMA)–ionomycin. We found that CD8αβ[+] γδ T cells constituted the majority of IFN-γ-producing γδ T cells in LNs, whereas they represented a minimal fraction of IL-17A producers (Fig. 2b), in agreement with their functional activities in the disease models above (Fig. 1c,f). To assess the molecular drivers of this IFN-γ production, we stimulated peripheral γδ T cells for 24 h either with IL-12 and IL-18, which are known to elicit IFN-γ secretion by innate-like T cells in the absence of TCR stimulation[18,19], or with TCR/CD28 agonists as used for activation of conventional adaptive-like T cells. This revealed that only IL-12/IL-18, but not TCR stimulation, triggered IFN-γ production by CD8αβ[+] γδ T cells (Fig. 2c), thus placing them as innate IFN-γ producers. We also assessed whether CD8αβ[+] γδ T cells can mount a CD16-mediated cytotoxic response similar to that seen in human CD8[+] γδ T cells[16]. However, we found little to no CD16 expression on CD8αβ[+] γδ T cells from LNs of WT mice (Extended Data Fig. 2a).

We then assessed whether the CD8αβ[+] γδ T cell phenotype represented a committed subset or a plastic cellular state. We stimulated in vitro, purified CFSE-labeled CD8αβ[+], CD8αα[+] and CD8[−] γδ T cells from pooled LNs/spleen of TCRα[−/−] mice (that are enriched for γδ T cells with normal functional potential), with a cytokine activation cocktail (containing IL-2, IL-4, IL-7 and IL-15) plus/minus TCR stimulation (that is, with anti-TCRδ antibody GL3). After a 3-day stimulation, we found substantial numbers of CD8αβ[+] γδ T cells only in cultures seeded with cells of this phenotype, in contrast to the vestigial CD8αβ[+] fractions derived from cultures of CD8αα[+] or CD8[−] γδ T cells (Extended Data Fig. 2b). To support these findings in vivo, we adoptively transferred purified CFSE-labeled CD8[−] or CD8αβ[+] γδ T cells isolated from TCRα[−/−] mice

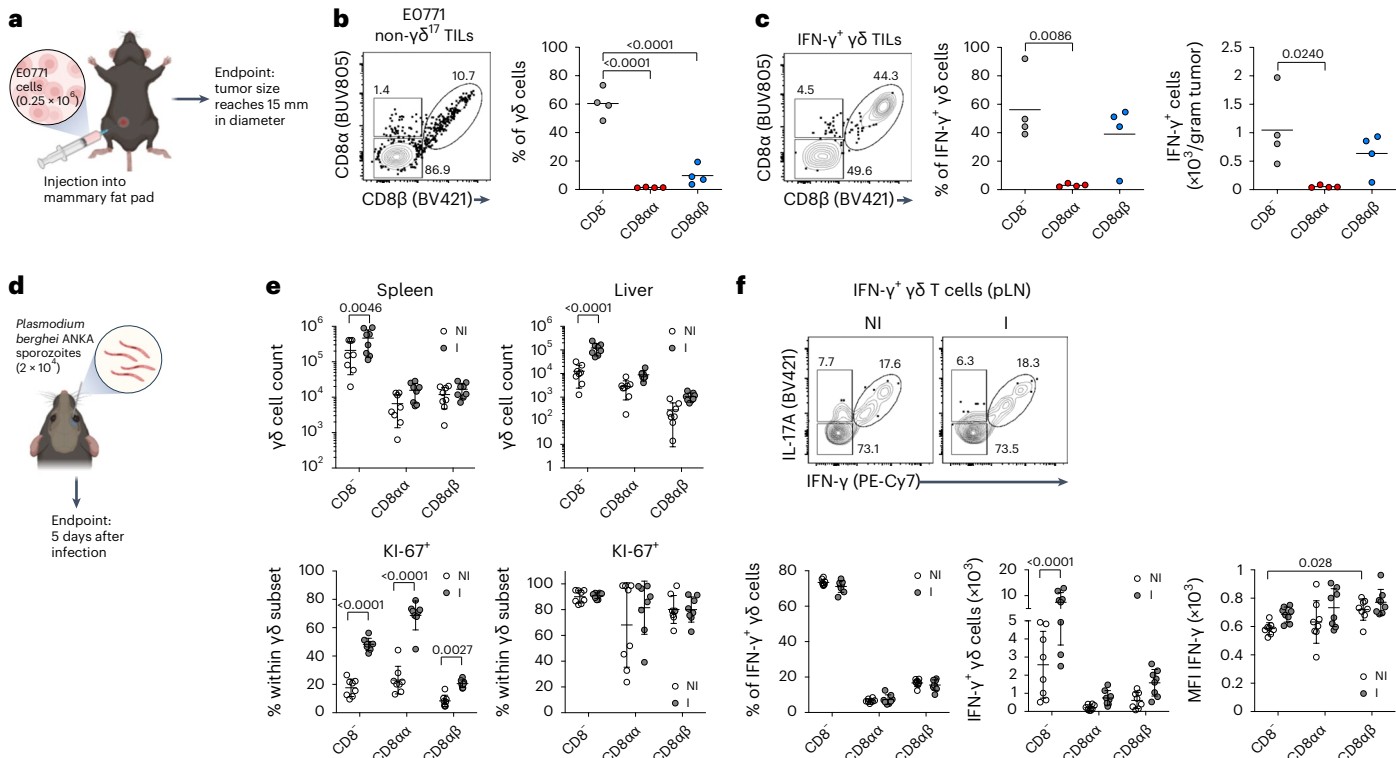

**Fig. 1 | CD8αβ⁺ γδ T cells respond to tumor challenge and malaria infection.**
**a**, Schematic of experimental design; E0771 tumor cells were injected into the mammary fat pad of B6 WT mice. Endpoint was considered when tumors reached 15 mm in diameter, at which point mice were culled and tumors and distal LNs were collected. Created with BioRender.com. **b**, Representative plot shows non-γδ[17] TILs isolated from tumor tissue. Cells were stained for CD8α/CD8β. The summary graph shows the frequency of CD8⁻, CD8αα⁺ and CD8αβ⁺ cells among γδ T cells in tumor (*n* = 4 mice). **c**, TILs from E0771 tumors were stimulated in vitro with PMA–ionomycin in the presence of brefeldin A for subsequent intracellular IFN-γ staining. Representative plot shows IFN-γ-expressing γδ TILs stained for CD8α/CD8β. Summary graphs show frequency or number of CD8⁻, CD8αα⁺ and CD8αβ⁺ cells among IFN-γ⁺ γδ T cells within the tumor. Tumor data are from one independent experiment (*n* = 4 mice). **d**, Schematic of experimental design; B6 WT mice were infected with *P. berghei* ANKA sporozoites via retro-orbital injection with $2 \times 10^4$ sporozoites. Mice were euthanized at day 5 after infection.

Spleen, liver and peripheral LNs were collected. Created with BioRender.com. **e**, Summary graphs show total cell counts of CD8⁻, CD8αα⁺ and CD8αβ⁺ γδ⁺ T cells in spleen and liver of noninfected (NI) and infected (I) mice. The percentage of Ki-67⁺ cells among CD8⁻, CD8αα⁺ and CD8αβ⁺ γδ T cells from spleen and liver is depicted (*n* = 8 mice). **f**, Lymphocytes from peripheral LNs (pLNs) of noninfected and infected mice were stimulated in vitro with PMA–ionomycin in the presence of brefeldin A. Representative plots show IFN-γ-expressing γδ⁺ T cells stained for CD8α/CD8β. Summary graphs show frequency or number of CD8⁻, CD8αα⁺ and CD8αβ⁺ cells among IFN-γ⁺ γδ T cells and their mean fluorescence intensity (MFI) values. Malaria data are from two independent experiments (*n* = 8 mice), and each dot represents an individual mouse. Percentages of gated cells are indicated in plots. Data are shown as the mean ± s.d. *P* values are indicated (one-way analysis of variance (ANOVA; **b** and **c**) or two-way ANOVA (**e** and **f**) with Sidak's or Tukey's test).

into immunodeficient *Rag2⁻/⁻γc⁻/⁻* recipients. After 3 days, we analyzed the lymphocytes retrieved from spleen. As expected, we observed lymphopenia-induced proliferation of cells in both transfers, but the phenotype of each subset was maintained (Fig. 2d). Furthermore, we validated the key finding that activated CD8⁻ γδ T cells from WT mice do not acquire CD8αβ expression, neither in vitro (Extended Data Fig. 2c), nor in vivo (Extended Data Fig. 2d). Collectively, these data demonstrate that CD8αβ expression characterizes a specific and stable subset of IFN-γ-producing γδ T cells that populates multiple peripheral organs from the perinatal period.

Since CD8αβ heterodimers are best recognized for their capacity to bind major histocompatibility complex (MHC) class I complexes, thus being required for the selection and antigen recognition of CD8⁺ αβ T cells, we investigated whether MHC class I expression was necessary for generation of CD8αβ⁺ γδ T cells. Importantly, we found expected numbers of CD8αβ⁺ γδ T cells in mice lacking the essential MHC class I component, β2-microglobulin β2m (Fig. 2e), which suggests that MHC class I is not necessary for CD8αβ expression on γδ T cells. Importantly, CD8αβ⁺ γδ T cells from LNs of β2m-deficient (β2m⁻/⁻) mice retained their effector phenotype and produced IFN-γ similarly to littermate controls upon stimulation in vitro with PMA–ionomycin

(Extended Data Fig. 2e). Also of note, in mice deficient in αβ T cells, that is, in either TCRβ⁻/⁻ or TCRα⁻/⁻ mice, there was an increase in the absolute numbers of CD8αβ⁺ γδ T cells (Fig. 2e and Extended Data Fig. 2f), which demonstrates they are not diverted from the αβ T cell pathway and raises the issue of their developmental trajectory in the thymus.

## CD8αβ⁺ γδ T cells develop in the perinatal thymus

Analysis of the thymus ex vivo along mouse ontogeny revealed that CD8αβ⁺ γδ T cells develop as a perinatal wave, appearing in the late fetal thymus and peaking just after birth (Fig. 3a). Consistent with the presence of CD8αβ⁺ γδ T cells in the periphery, newborn β2m⁻/⁻ mice had normal CD8αβ⁺ γδ T cell levels in the thymus (Extended Data Fig. 3a). To begin to better understand their developmental requirements, we built on our previous observation that pharmacological inhibition of PI3K signaling augmented development of CD8αβ⁺ γδ T cells[12]. Consistent with this, CD8αβ⁺ γδ T cells, expressing Ly6a (Sca-1), were also significantly increased in 8-day fetal thymic organ cultures (FTOCs) of embryonic day-15 thymic lobes from PI3K p110δ-deficient animals (Fig. 3b). By contrast, CD8αβ⁺ γδ T cells were dramatically reduced in 8-day FTOCs from mice expressing constitutively active p110δ^E1020K (Fig. 3c), which could be rescued by supplementation of a pan-PI3K

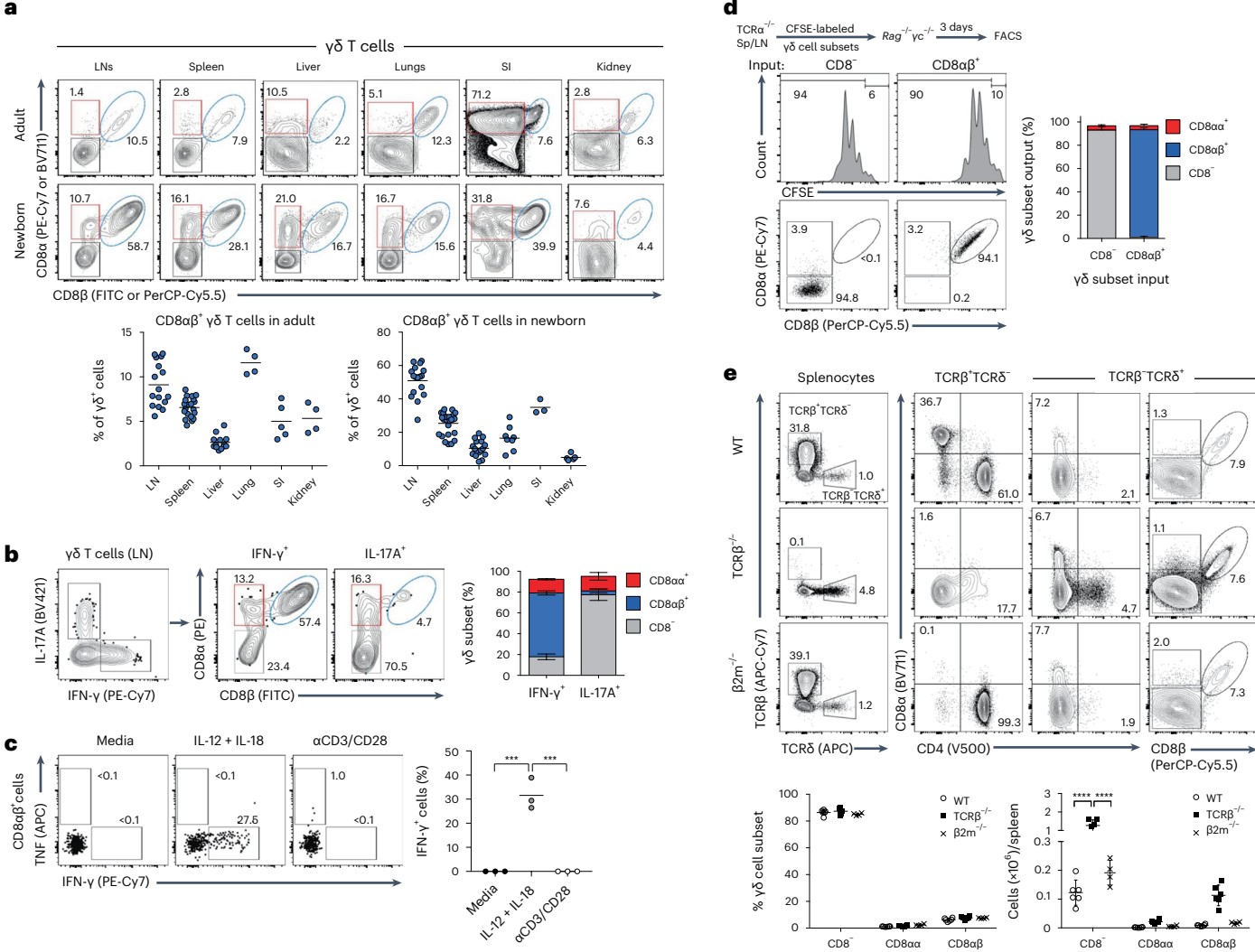

**Fig. 2 | CD8αβ⁺ γδ T cells are a stable IFN-γ-producing subset. a**, Representative plots show γδ T cells from indicated peripheral tissues of adult and neonatal B6 WT mice. Cells were stained for CD8α/CD8β. Data are representative of one to eight independent experiments (*n* = 3–20 mice). Summary graphs depict the relative contribution of CD8αβ⁺ cells to the γδ T cell compartment in tissues of adult and newborn B6 WT mice. Each symbol represents an individual mouse. SI, small intestine. **b**, Representative plots show intracellular IFN-γ and IL-17A in γδ T cells from neonatal B6 WT LNs. IFN-γ⁺ and IL-17A⁺ γδ T cells were analyzed for CD8α/CD8β expression. Summary graph of two independent experiments (*n* = 6 mice) showing percentages of CD8⁻, CD8αβ⁺ and CD8αα⁺ among IFN-γ⁺ and IL-17⁺ γδ T cells. **c**, Enriched γδ T cells from B6 WT LNs were cultured for 24 h in media alone, in IL-12 plus IL-18 (100 ng ml⁻¹ each) or on plate-bound anti-CD3ε plus soluble anti-CD28 (2 µg ml⁻¹ each). Representative plots show CD8αβ⁺ non-γδ¹⁷ cells stained for intracellular TNF/IFN-γ. The summary graph shows the percentage of IFN-γ⁺ cells among CD8αβ⁺ cells. Data are representative of two

independent experiments. Each symbol represents a triplicate. \*\*\**P* = 0.0001. **d**, CD8⁻ and CD8αβ⁺ γδ T cells purified from spleen and LNs of TCRα⁻/⁻ mice, labeled with CFSE and injected into *Rag2*⁻/⁻*γc*⁻/⁻ recipients. After 3 days, proliferation of donor cells from recipient spleens was analyzed. Representative histograms and plots showing CFSE dilution and CD8α/CD8β expression on donor cells. The summary graph depicts composition of recovered donor cells (*n* = 5–7 recipients). **e**, Lymphocytes from B6 WT, TCRβ⁻/⁻ and β2m⁻/⁻ spleens stained for TCRβ, TCRδ, CD4, CD8α and CD8β. Representative plots show CD4, CD8α and CD8β expression on αβ and γδ T cells. Data are representative of two independent experiments (*n* = 4–6 mice). Summary graphs depict the relative contribution of CD8⁻, CD8αβ⁺ and CD8αα⁺ cells to the γδ T cell compartment and total cell counts in adult mice. Each symbol represents an individual mouse. Percentages of gated cells are indicated. Error bars show the mean ± s.d. *P* values are indicated (one-way ANOVA (**c**) or two-way ANOVA (**e**) with Tukey's, Dunnett's or Sidak's test). FACS, fluorescence-activated cell sorting.

---

inhibitor (Extended Data Fig. 3b). CD8αβ⁺ γδ T cells from 8-day WT FTOCs were present in γδ²⁴⁺ (early CD24-expressing γδ progenitors), γδ^TN (CD24⁻CD44⁻CD45RB⁻ triple-negative) and γδ^RB (CD45RB⁺) cell populations, but unexpectedly not in the two thymic CD24⁻ subsets previously considered to represent terminally differentiated γδ T cells; the CD44⁺CD45RB⁻ γδ¹⁷ subset, and the CD44⁺CD45RB⁺ γδ^RB/44 subset (which we had previously considered as mature γδ^IFN cells[10]; Fig. 3d). Indeed, in 14-day WT FTOCs, CD8αβ⁺ γδ T cells accumulated only in the γδ^RB population (Extended Data Fig. 3c). Unlike early embryonically derived γδ T cell subsets (for example, DETC precursors), CD8αβ⁺ γδ T cells utilize a broad selection of Vγ regions with little suggestion of

TCRγδ restriction (Fig. 3e and Extended Data Fig. 3d), and do not display features of NKT-like γδ T cells[5], for example, expression of NK1.1 (Extended Data Fig. 3e).

To gain a more global understanding of CD8αβ⁺ γδ T cells, and to probe their developmental characteristics, we performed single-cell RNA-sequencing (RNA-seq) analysis on sorted γδ thymocytes from 8-day WT FTOC in media alone or with pan-PI3K inhibitor (in which CD8αβ⁺ γδ T cells are greatly expanded). A uniform manifold approximation and projection (UMAP) plot of the combined cells displayed eight clusters (Fig. 3f). Cluster 6 displayed features associated with very early stages of thymocyte development (for example, *Bcl11b, Cd24a,*

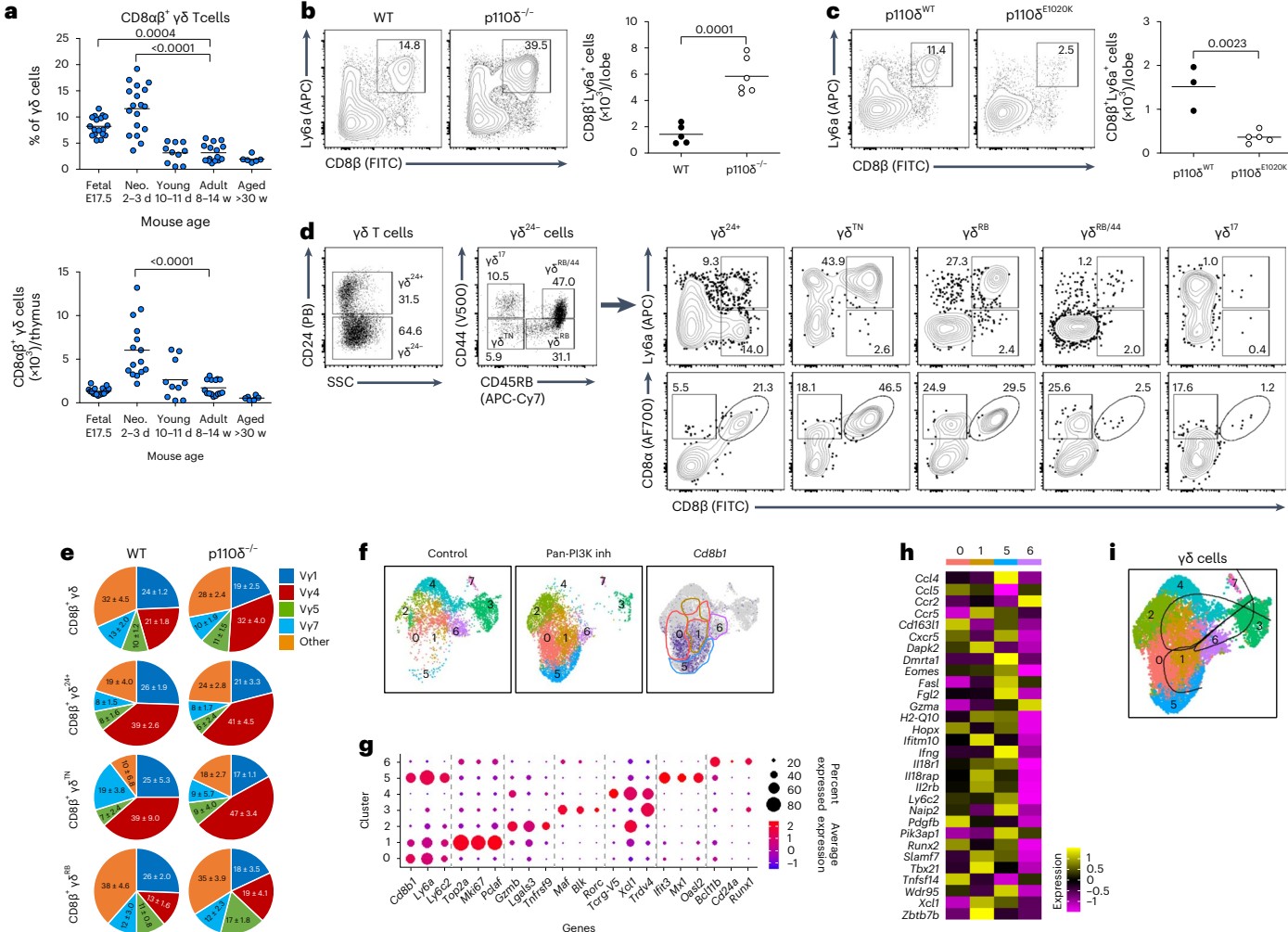

**Fig. 3 | CD8αβ⁺ γδ T cells develop from a discrete perinatal thymic wave.**
**a**, Summary graphs of relative contribution to (upper graph) and total numbers of (lower graph) CD8αβ⁺ cells in thymic γδ T cell compartment in B6 WT mice during ontogeny. Embryonic day-17.5 (fetal), day-2/3 (neonates), day-10/11 (young), week-4–14 (adult) and >30-week-old (aged) mice were pooled from 2–5 experiments (n = 6–18). Each symbol represents an individual mouse. **b,c**, Representative plots show Ly6a/CD8β on non-γδ[17] T cells from WT (n = 5) and p110δ[−/−] (n = 6) (**b**) or p110δ[WT] (n = 3) and p110δ[E1020K] (n = 5) (**c**) mice, after 8-day FTOC. Summary graphs show absolute number of CD8β⁺Ly6a⁺ γδ T cells. Each symbol represents two thymic lobes pooled. Data are pooled from (**b**) or representative of (**c**) two independent experiments. **d**, Representative flow cytometry plots of γδ T cells from 8-day WT FTOC showing total γδ T cells, CD24⁻ γδ (γδ[24−]) T cells, and indicated γδ subsets stained for Ly6a/CD8β and CD8α/ CD8β. Data are representative of at least two independent experiments. **e**, Pie charts depict Vγ usage by CD8αβ⁺ γδ T cells in distinct γδ subsets from WT and

p110δ[−/−] 8-day FTOC. Data are from one independent experiment. **f**, UMAP plots show γδ T cells from pan-PI3K inhibitor (ZSTK474; 0.25 μM) treated or untreated 8-day WT FTOC. Feature plot showing expression of *Cd8b1* in γδ T cells from untreated and treated FTOC combined. **g**, Dot plot showing scaled expression of most differentially expressed genes (adjusted P ≤ 0.05), scaled across all clusters. Dot size represents the percentage of cells in each cluster. **h**, Heat map shows differentially expressed genes between indicated clusters within integrated γδ T cells from PI3K inhibitor-treated and untreated 8-day WT FTOC. **i**, Pseudotemporal trajectories computed using Slingshot trajectory inference R package. Cells in cluster 6 were provided as the root node. Percentages of gated cells are indicated. TN is triple negative for CD24, CD44 and CD45RB; RB is CD45RB⁺; RB/44 is double positive for CD45RB and CD44. Error bars show the mean ± s.d., P values are indicated (one-way ANOVA with Dunn's or Dunnett's test (**a**) or unpaired two-tailed Student's t-test (**b** and **c**)).

*and Runx1* expression), cluster 3 expressed genes associated with γδ[17] cells (for example, *Maf*, *Blk* and *Rorc*), cluster 4 expressed Vγ5 and Vδ1 transcripts, and cluster 1 contained predominantly cycling cells (Fig. 3g and Extended Data Fig. 3f). Of the three *Cd8b1*⁺ clusters (clusters 0, 1 and 5), cluster 5 expressed genes associated with a mature γδ T cell phenotype (for example, *Ifng*) and, notably, displayed a signature that correlated closely with previously identified innate-like memory T cell programs (for example, *Eomes*, *Ly6c2* and *Cxcr5*)[20] (Fig. 3h). We validated some key innate-like signature markers—CD8β, Ly6a and Eomes—at the protein level, and found them to be maintained throughout mouse ontogeny (Extended Data Fig. 4). Interestingly, cluster 5 also displayed a strong signature of type I interferon signaling (for example, *Ifit3*, *Mx1* and *Oasl2*), and by slingshot trajectory analysis appeared to

represent the most terminally differentiated subset in a CD8αβ⁺ γδ T cell-specific pathway distinct from that used by other cells of the IFN-γ pathway, for example, Vγ5⁺Vδ1⁺ γδ T cells, which were found in cluster 4 (Fig. 3i). Thus, these data characterize CD8αβ⁺ γδ T cells as a distinct innate-like T cell population with IFN-γ-secreting potential, which develops from a discrete perinatal thymic wave and displays a diverse TCRγδ repertoire and a strong type I interferon-responsive gene signature.

## IL-4, IL-7 and low TCR signaling favor CD8αβ⁺ γδ T cell development
We next assessed the developmental requirements of CD8αβ⁺ γδ T cells. To focus only on the IFN-γ pathway, we set up 8-day FTOCs from

RORγt-GFP mice[21], which allowed us to remove RORγt[+] (that is, GFP[+]) γδ[17] cells and γδ[17] progenitors from our analyses. Entry of γδ progenitors into the IFN-γ pathway has long been associated with strong signaling through TCRγδ[4,10,22,23], as indicated by robust upregulation of CD73 (ref. 24). Such strong signaling also correlates with rapid upregulation of CD45RB; for example, as seen with precursors of Vγ5[+]Vδ1[+] DETCs that express CD45RB as early as the γδ[24+] stage[3].

After gating on RORγt[–] cells, we compared CD8αβ[+] γδ T cells to CD8αα[+] γδ T cells, as an example of a previously well-characterized IFN-γ-secreting subset[25]. CD73 and CD45RB were notably lower in CD8αβ[+] γδ T cells than in CD8αα[+] γδ T cells, suggesting that they had received a weaker TCRγδ signal during development (Fig. 4a). Interestingly, this lower CD73 expression in CD8αβ[+] γδ T cells correlated with much higher expression of CD5 that is associated with other innate-like lymphocyte subsets, for example, B1 B cells[26] (note that CD5 expression does not correlate with TCR signal strength, as it does for mature αβ T cells, during γδ T cell development[10]; Fig. 4b).

To directly assess the role of TCRγδ signaling on CD8αβ[+] γδ T cell development, a series of 8-day WT FTOCs were established in which TCRγδ signal strength was either increased using the agonist TCRδ antibody GL3 or decreased with the MEK1/2 inhibitor UO126. Consistent with lower CD73 (and CD45RB) expression in developing CD8αβ[+] γδ T cells, curtailing TCRγδ signal strength increased the proportion of CD8αβ[+] γδ T cells that developed in these cultures (Fig. 4c). By contrast, augmenting TCRγδ signal strength decreased the proportion of CD8αβ[+] γδ T cells that were generated (Fig. 4c). We next assessed whether reduced TCRγδ signal strength was also sufficient to upregulate the key innate-like transcription factor Eomes. However, although addition of UO126 to 8-day WT FTOCs increased the absolute number of CD8αβ[+] γδ T cells, these cells expressed marginally less Eomes (Fig. 4d,e). Innate-like αβ T cells were previously shown to upregulate Eomes in response to IL-4 (ref. 27). Consistent with this, addition of IL-4 alone to 8-day WT FTOCs increased Eomes expression in CD8αβ[+] γδ T cells but did not increase their absolute cell number. However, when IL-4 was combined with reduced TCRγδ signal strength using UO126, we observed ~10-fold increase in CD8αβ[+] γδ T cells, a majority of which had significantly upregulated Eomes expression (Fig. 4d,e). A similar synergistic effect was observed when IL-4 was combined with a pan-PI3K inhibitor as an alternative means to reduce TCRγδ signal strength (Extended Data Fig. 5a). Conversely, using a loss-of-function approach, we found that the CD8αβ[+] γδ T cell compartment was reduced in IL-4Rα[–/–] mice (Extended Data Fig. 5b).

To complement these analyses, we also considered the role of IL-7, as it had been reported to expand fetal γδ thymocytes expressing CD8αβ heterodimers[14]; and we had found that inhibition of PI3K upregulated IL-7R expression on developing γδ T cells[11,13,22]. In both 8-day WT FTOCs and in neonates, higher IL-7R levels were detected on CD8αβ[+] γδ T cells, when compared with either CD8αα[+] or CD8β[–] γδ T cells (Fig. 4f and Extended Data Fig. 5c). Consistent with this, IL-7 alone was able to substantially increase the percentage and absolute numbers of CD8αβ[+] γδ T cells in 8-day WT FTOC, but did not upregulate Eomes (Fig. 4d,e). Although addition of the MEK1/2 inhibitor UO126 alone failed to rescue this effect on Eomes, the combination of IL-7, UO126 and IL-4 provided the largest yield of Eomes[+] CD8αβ[+] γδ T cells in our assays (Fig. 4d,e). Furthermore, the addition of IL-7 rescued the striking impact of the agonist TCRδ antibody GL3 on CD8αβ[+] γδ T cell numbers (Fig. 4g). Collectively, these data demonstrate that CD8αβ[+] γδ T cell development is promoted by attenuated/weaker TCRγδ signal strength together with two critical cytokines: IL-4 as the main inducer of the Eomes[+] innate-like phenotype, and IL-7 as the major driver of CD8αβ[+] γδ thymocyte expansion.

## IL-7R signaling drives malignant CD8αβ[+] γδ T cell generation

We have previously shown that, in addition to supporting normal T cell differentiation, IL-7–IL-7R signaling promotes the development of lymphoid malignancies[28–30]. To investigate the potential relationship between IL-7–IL-7R signaling and CD8αβ[+] γδ T cells in the context of cancer, we used transgenic mice expressing a frequent human (h) oncogenic driver mutation in the IL-7R effector and key signal transducer STAT5B, Asp642His (N642H), which is associated with poor prognosis and increased risk of relapse[31]. Upon expression under the control of the Vav1 promoter, hSTAT5B[N642H] triggered leukemia or lymphoma development, which manifested as a transplantable CD8[+] T cell disease[32]. Interestingly, hSTAT5B[N642H]-transformed γδ T cells were shown to reconstitute disease with pathological characteristics similar to those observed in participants with hepatosplenic T cell lymphoma[33], an aggressive malignancy of γδ T cell origin primarily affecting the liver and the spleen[34]. We, therefore, established FTOCs using embryonic day (E) 15 thymic lobes from Vav1.hSTAT5B[N642H] mice, and observed augmented proportions (2.5-fold) and absolute numbers (5.7-fold) of CD8αβ[+] γδ T cells compared to non-transgenic littermate control FTOCs (Fig. 5a and Extended Data Fig. 6a). Similarly, the analysis ex vivo of thymi from Vav1.hSTAT5B[N642H] mice showed a sizable (~74-fold) increase in CD8αβ[+] γδ T cell numbers, whereas CD8[–] and CD8αα[+] γδ T cells increased only ~2-fold and ~10-fold, respectively (Fig. 5b). Vav1.hSTAT5B[N642H] thymi presented an atypical accumulation of mature CD24[–] γδ T cells (Extended Data Fig. 6b), and CD8αβ[+] γδ T cells were substantially more enriched (~35-fold versus ~4-fold) in the mature CD24[–] than in the immature CD24[+] compartment (Extended Data Fig. 6c). In the periphery, Vav1.hSTAT5B[N642H] mice presented an accumulation of γδ T cells in the spleen (Extended Data Fig. 6d), with a marked increase in CD8αβ[+] γδ T cell numbers (Fig. 5c), and a very clear enrichment for CD8αβ[+] γδ T cells at the expense of their CD8[–] counterparts (Extended Data Fig. 6e), which was also observed in the LNs (Extended Data Fig. 6f). These results firmly establish that oncogenic STAT5B signaling promotes the thymic development and expansion of CD8αβ[+] γδ T cells that seed and accumulate in peripheral lymphoid organs of this aggressive hematological disease. Strikingly, ~95% of Vav1.hSTAT5B[N642H] transgenic CD8αβ[+] γδ T cells produced IFN-γ (compared to ~40% of controls) (Extended Data Fig. 6g), suggesting the targeted dysregulation of this perinatal IFN-γ-biased γδ T cell subset.

To complement this, we used Rosa26-hIL-7R.huCD2-Cre mice that overexpress human IL-7R. We recently characterized these mice as a model of IL-7R-mediated T cell acute T-ALL, since they displayed a proliferative thymic phenotype, increased thymus size and tumor cell dissemination to many tissues, including the bone marrow and the spleen, leading to splenomegaly[35]. Interestingly, the thymocytes expanded in this model presented a CD8[+]CD4[–] phenotype but failed to stain for TCRβ[35]. We found that the vast majority (>90%) of thymocytes and peripheral LN cells from these animals were positive for TCRδ (Fig. 5d), and >80% of these γδ T cell leukemic blasts expressed CD8αβ heterodimers (Fig. 5e,f). These data clearly demonstrate that IL-7–IL-7R signaling is a major promoter of malignant CD8αβ[+] γδ T cell development/expansion in this preclinical model of T-ALL.

## CD8β expression defines a subset of human γδ T-ALL

Finally, we investigated whether CD8β might constitute a useful marker in the clinical T-ALL setting, which seemed a particularly interesting hypothesis given the perinatal origin of CD8αβ[+] γδ T cells and the pediatric nature of a majority of T-ALL cases[36]. We analyzed RNA-seq data from 20 diagnostic γδ T-ALL samples from the St. Jude's Children Research Hospital (Memphis, USA) patient cohort and identified three cases with high CD8B, intermediate CD8A and low/intermediate CD4 transcript levels (Fig. 6a). In agreement with our findings in the mouse, they all expressed high IL7R (Fig. 6a). This is a clear indication of the involvement of aberrant IL-7–IL-7R signaling in human CD8αβ[+] γδ T-ALL, given that high levels of IL7R in patients with T-ALL are known to associate with oncogenic IL-7R-dependent signaling activation[35]. Two of the CD8B[hi] cases belonged to the HOXA subgroup (with either DDX3X–MLLT10 or KMT2A–AFDN gene fusions) and expressed a Vγ9Vδ1 TCR (Extended

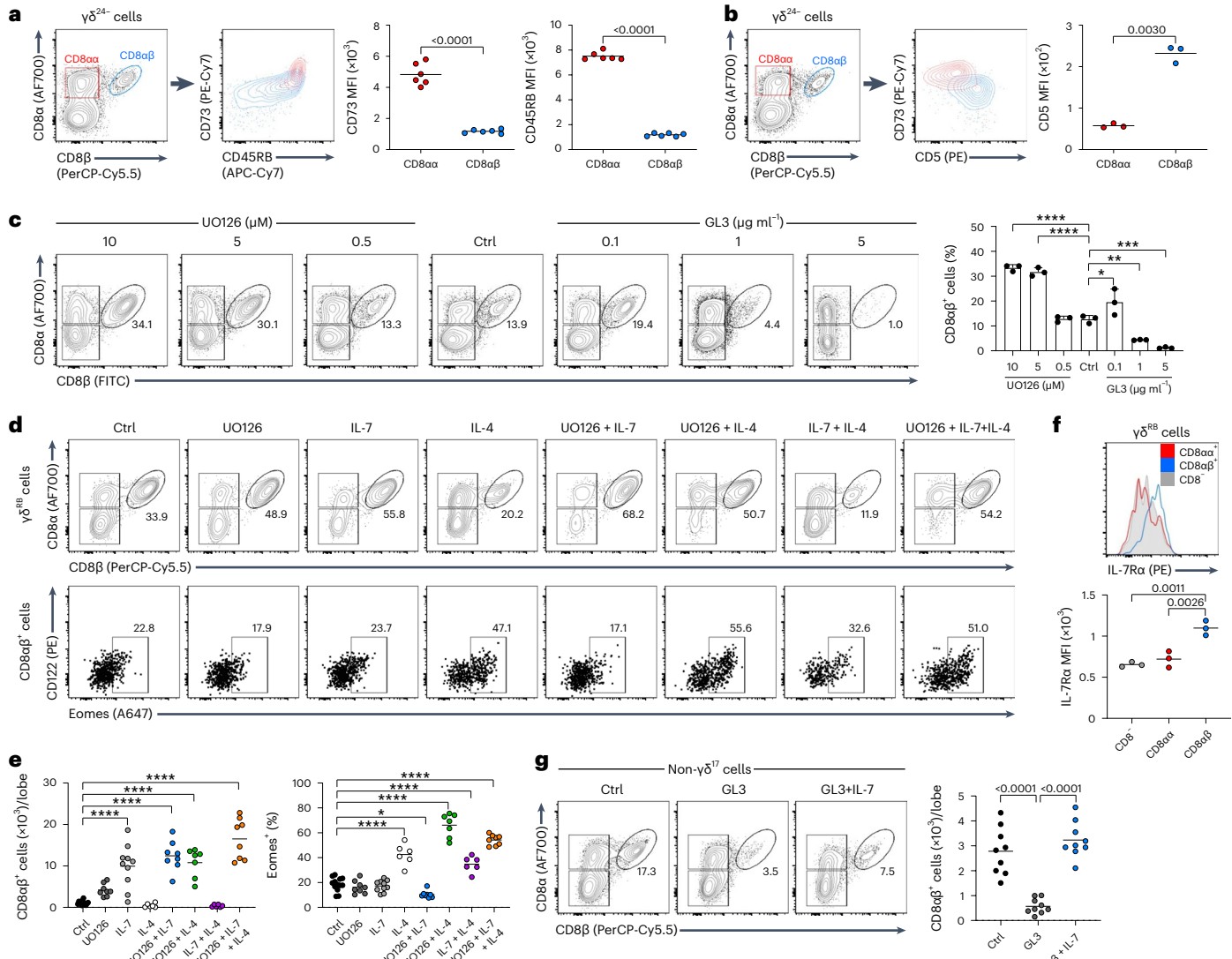

**Fig. 4 | IL-4, IL-7 and low TCR signaling promote CD8αβ⁺ γδ T cell development. a,b**, RORγt⁻ or WT (respectively) γδ²⁴⁻ T cells from 8-day FTOC with CD8αα⁺ (red) and CD8αβ⁺ (blue) cells overlaid for CD73 and CD45RB expression (**a**) or CD73 and CD5 expression (**b**), respectively. Summary graphs show MFI values of CD73 and CD45RB (**a**) or CD5 (**b**). Each symbol represents two thymic lobes pooled (n = 6 pairs). Data are representative of two independent experiments. **c**, Representative plots show non-γδ¹⁷ T cells from 8-day WT FTOC stained for CD8α/CD8β in the presence of MEK1/2 inhibitor UO126, TCRδ-activating antibody GL3, or under control conditions (Ctrl). The summary graph shows the percentage of CD8αβ⁺ cells within non-γδ¹⁷ cells (n = 3 samples per group). Data are representative of two independent experiments. *P = 0.0109, **P = 0.0025, ***P = 0.0001 and ****P < 0.0001. **d**, Representative plots show CD8αβ⁺ γδ^RB cells from 8-day WT FTOCs (top) stained for CD122/Eomes (bottom) in the presence of UO126 (5 μM), IL-7 (10 ng ml⁻¹) and/or IL-4 (20 ng ml⁻¹), or under control conditions. **e**, Summary graphs show absolute numbers of

CD8αβ⁺ cells (left), or the percentage of Eomes⁺ CD8αβ⁺ cells (right; Ctrl n = 13, UO126 n = 8, IL-7 n = 10, IL-4 n = 6, UO126 + IL-7 n = 8, UO126 + IL-4 n = 7, IL-7 + IL-4 n = 6, UO126 + IL-7 + IL-4 n = 8 samples). Data are pooled from five independent experiments. *P = 0.0206, ****P < 0.0001. **f**, Representative histogram shows IL-7Rα expression on CD8αα⁺ (red), CD8αβ⁺ (blue) and CD8⁻ (gray) cells among γδ^RB cells from 8-day WT FTOC. The summary graph shows the MFI of IL-7Rα (n = 3 samples). Data are representative of two independent experiments. **g**, Representative plots show non-γδ¹⁷ T cells from 8-day WT FTOCs stained for CD8α/CD8β plus GL3 (1 μg ml⁻¹), GL3 plus IL-7 (10 ng ml⁻¹) or under control conditions. Summary graphs show absolute numbers of CD8αβ⁺ cells (n = 9 samples per group). Data are pooled from three independent experiments. Each symbol in the summary graphs represents at least three lobes pooled. Data are shown as the mean ± s.d. P values are indicated (unpaired two-tailed Student's t-test (**a** and **b**) or one-way ANOVA with Tukey's or Dunnett's test (**c** and **e**–**g**)).

Data Table 1). The remaining case was an immature T-ALL with a Vγ8Vδ1 TCR (Extended Data Table 1). We next evaluated by flow cytometry 37 primary and 5 patient-derived xenograft γδ T-ALL samples from a Hôpital Necker Enfants-Malades (Paris, France) cohort, identifying two cases with CD8β surface expression (Fig. 6b), one of which overexpressed TLX3. Taken together, our observations suggest that 5–15% of patients with γδ T-ALL are CD8⁺ (Fig. 6c). We thus propose CD8β as a new marker to include in the classification of patients with γδ T-ALL, and urge the phenotypic assessment of its expression within human γδ T cells in other leukemia and lymphoma cohorts, as well as in other diseases.

## Discussion

A striking characteristic of γδ T cell biology are the temporally regulated developmental 'waves' that egress the thymus and populate specific peripheral tissues[1]. These initiate in the embryo and generate the earliest T cells found in the mouse, which notably include DETCs expressing an invariant Vγ5Vδ1 TCR and displaying tissue-repair properties and antitumor cytotoxicity, but limited capacity to produce IFN-γ or IL-17A; and Vγ6⁺ (typically also Vδ1⁺) γδ T cells highly biased toward IL-17A production, that populate multiple tissues such as the dermis, tongue, testis, uterus, lung, adipose tissue and the brain meninges[1]. Here, we

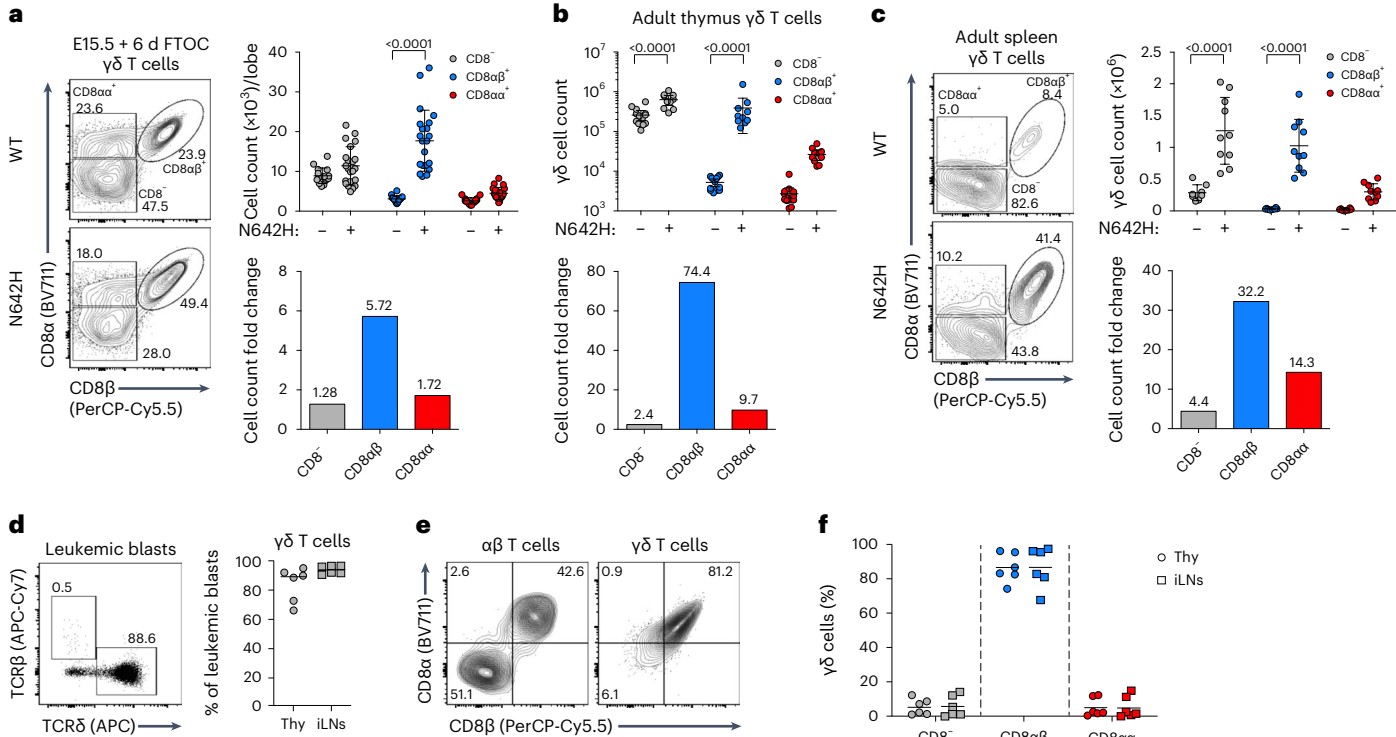

**Fig. 5 | IL-7R–STAT5B signaling drives malignant CD8αβ⁺ γδ T cell development. a**, Analysis of 6-day FTOCs from *Vav1*.hSTAT5B^N642H (N642H) and STAT5B^WT (WT) littermate controls. Representative plots show CD8α and CD8β expression on γδ T cells. The summary graph depicts total cell numbers of CD8⁻, CD8αβ⁺ and CD8αα⁺ γδ T cells. Mean fold changes of CD8⁻, CD8αβ⁺ and CD8αα⁺ γδ T cell counts between WT and N642H are depicted in bar graph with fold change indicated above each bar. Data are from two independent experiments. Each symbol represents one lobe (*n* = 16–22 lobes). **b**, The summary graph depicts total cell numbers of CD8⁻, CD8αβ⁺ and CD8αα⁺ γδ T cells in adult thymus from N642H and WT littermate controls. Mean fold changes of CD8⁻, CD8αβ⁺ and CD8αα⁺ γδ T cell counts between WT and N642H are depicted in bar graph with fold change indicated above each bar. Data are from two independent experiments. Each symbol represents an individual mouse (*n* = 10–13 mice). **c**, Representative plots show CD8α and CD8β expression on γδ⁺ T cells from adult spleen of N642H and WT littermate controls. Summary graphs depict total cell

numbers of CD8⁻, CD8αβ⁺ and CD8αα⁺ γδ T cells. Mean fold changes of CD8⁻, CD8αβ⁺ and CD8αα⁺ γδ⁺ T cell counts between WT and N642H are depicted in the bar graph with fold change indicated above each bar. Data are from two independent experiments. Each symbol represents one mouse (*n* = 8–10 mice). **d–f**, Analysis of Rosa26-hIL-7R.huCD2-Cre transgenic mice. Data are representative of two independent experiments (*n* = 6 mice). **d**, Representative flow cytometry plot shows GFP⁺-gated leukemic blasts from transgenic animals stained for TCRβ and TCRδ. The summary graph presents the percentage of γδ T cells in thymus and inguinal LNs (iLNs). **e**, Representative plots show expression of CD8α and CD8β on αβ and on γδ T cells. **f**, Summary plot of the percentages of CD8⁻, CD8αβ⁺ and CD8αα⁺ γδ T cells in thymus and peripheral LNs of transgenic mice. Each symbol represents an individual mouse. Data are shown as the mean ± s.d. *P* values are indicated (two-way ANOVA with Sidak's multiple-comparison test (**a–c**)).

---

additionally describe a perinatal IFN-γ-producing γδ T cell subset, which, unlike the aforementioned populations, expresses a diverse set of TCR Vγ chains (especially Vγ1, Vγ2 and Vγ4, in comparable proportions), and CD8αβ heterodimers as their unique phenotypic signature (among γδ T cells). Importantly, CD8αβ⁺ γδ T cells robustly produce IFN-γ after stimulation with IL-12/IL-18, rather than by TCR-dependent activation, a feature that denotes innate-like T cell behavior[19,37]. This is consistent with their IL-4-dependent thymic upregulation of Eomes[27], a key transcription factor for the memory-associated transcriptional program observed in innate CD8αβ⁺ αβ T cells[20,38]. Indeed, CD8αβ⁺ γδ T cells share a similar innate T cell transcriptomic signature, which includes expression of genes such as *Ifng*, *Runx2*, *Ly6c2* and *Fasl*[20,39] and defines a cellular phenotype maintained from birth into adulthood.

Our data also demonstrate that CD8αβ⁺ γδ T cells develop along a distinct trajectory in the perinatal thymus, characterized by delayed entry into the CD45RB⁺ IFN-γ pathway and lower surface levels of CD73. This reduced CD73 expression highlights a requirement for weaker TCRγδ signal strength for the optimal development of CD8αβ⁺ γδ T cells, a feature also previously noted for innate-like CD8αβ⁺ αβ T cells[40]. This firmly segregates CD8αβ⁺ γδ T cells from other CD45RB⁺ γδ T cells (for example, DETCs and Vγ1⁺ NKT-like γδ T cells), for which TCR–ligand engagement and strong TCRγδ signaling

have been developmentally implicated[3,5]. These distinct TCR signal strength-dependent developmental fates may be controlled via graded expression of TCR signaling-induced IRF4, which promotes effector T cell gene programs but suppresses TCF1-induced genes that are associated with the memory-like features of innate T cells[41]. Indeed, intriguingly, this requirement for lower TCRγδ signal strength aligns CD8αβ⁺ γδ T cells toward perinatal waves of γδ[17] cells, which also require attenuated TCRγδ signaling for optimal thymic development[4,10,12]. Nonetheless, the circumstances that provide this weaker TCRγδ signaling, and the implications that this has for subsequent TCRγδ specificity and repertoires, and for γδ T cell function, require further clarification.

The prompt secretion of IFN-γ likely underlies the functional relevance of perinatal CD8αβ⁺ γδ T cells, especially as these constitute the main γδ T cell IFN-γ source in newborn LNs, and still a sizeable fraction in adult LNs, while also being present in multiple other tissues. Many studies have shown that IFN-γ is the key effector cytokine produced by γδ T cells in the context of antiviral[42] and antitumor responses[2]. Here we found CD8αβ⁺ γδ T cells to be strong responders to both malaria infection and tumor challenge. In particular, they accounted for almost half the IFN-γ-producing γδ T cells within E0771 breast lesions, which represented a striking enrichment compared to distal LNs. As a limitation

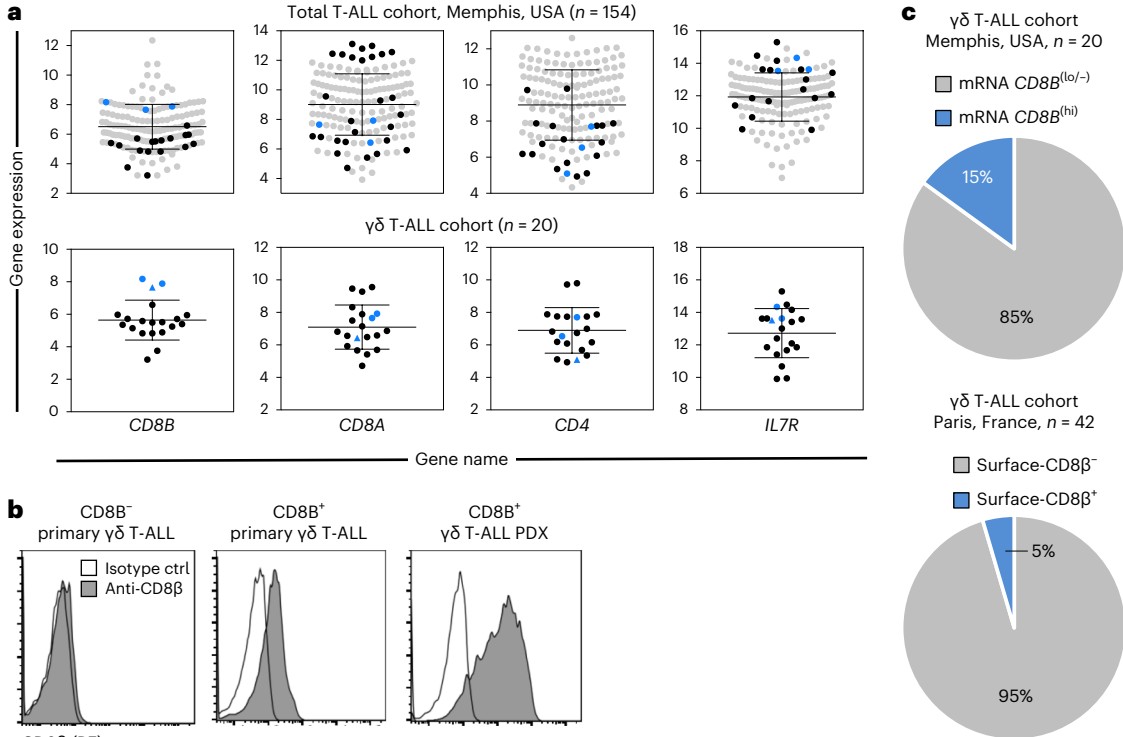

**Fig. 6 | CD8β expression defines a subset of human γδ T-ALL. a**, A T-ALL patient cohort (Memphis, USA) was analyzed by RNA-seq. Gene expression data of *CD8B*, *CD8A*, *CD4* and *IL7R* are shown. Each symbol depicts one patient with γδ T-ALL (*n* = 154). Dependent on their *CD8B* gene expression levels, patients with γδ T-ALL were subdivided into CD8B(lo/−) (black) or CD8B(hi) (blue). In the lower row, a zoomed-in view of all γδ T-ALL patient data is presented. CD8α flow cytometry confirmed surface-CD8 expression for two CD8B(hi) patients (blue dots), while one patient was surface-CD8α negative (blue triangle). Gene expression data are shown as the mean ± s.d. **b**, From a second γδ T-ALL patient cohort (Paris, France), primary samples and patient-derived xenografts (PDX) were analyzed by flow cytometry for surface-CD8β (*n* = 40). Representative histograms are depicted. **c**, Pie charts summarize CD8B and CD8β analyses of the two γδ T-ALL patient cohorts from **a** and **b**.

to our study, we currently lack the tools to selectively deplete CD8αβ⁺ γδ T cells and thereby assess their (non)redundant contributions to disease models; this will be a goal for follow-up research.

In humans, CD8αβ⁺ γδ T cells have been reported in various severe disease conditions, namely inflammatory bowel disease, melanoma, cardiovascular disease and chronic tuberculosis. Consistent with the functional potential of their mouse equivalents, human CD8αβ⁺ γδ T cells were shown to be endowed with 'NK-like' cytotoxicity and production of IFN-γ[15,16]. Moreover, a recent study analyzing human pediatric thymi described γδ T cells expressing *Cd8a* and *CD8b* within a 'type 1 cytotoxic' RNA-seq cluster[43]. This is especially interesting since neonatal human γδ T cells were found to be potent producers of IFN-γ, in stark contrast with their neonatal αβ T cell counterparts[44]. In fact, the expansion and type 1 effector differentiation of γδ T cells in utero upon both cytomegalovirus (CMV) and toxoplasma infections has been well documented[45,46]. Provocatively, another recent report described a sizable enrichment of CD8⁺ cells (displaying a terminal effector memory phenotype) among γδ T cells in CMV⁺ (compared to CMV⁻) grafts used in hematopoietic stem cell transplantation[47]. Unfortunately, however, the antibody used (SK1) was specific for CD8α, preventing confirmation of a selective expansion of CD8αβ⁺ γδ T cells in these human bone marrow samples. We expect the present study to encourage the assessment of CD8β expression within γδ T cells in human disease settings, building on our findings on T-ALL, to which we were drawn by the key role of IL-7–IL-7R in CD8αβ⁺ γδ T cell development and expansion. IL-7–IL-7R-mediated signaling (due to IL-7R gain-of-function mutations, IL-7R overexpression or microenvironmental IL-7) drives malignant T cell development and expansion in vivo[28,29]. Physiologically, IL-7 is also a critical factor for early γδ T cell development[48]. This, together

with our observations noting the high levels of IL-7Rα expression on CD8αβ⁺ γδ T cells and their marked expansion by IL-7 in FTOCs, led us to focus on the malignant potential of CD8αβ⁺ γδ T cells. The data obtained in mouse models herein, and from human T-ALL samples, demonstrated that high expression of WT IL-7Rα can contribute to T-ALL development, even in the absence of mutational activation of the receptor[35]. Here, using the Rosa26-hIL-7R.huCD2-Cre model, where *IL7R* is conditionally expressed in lymphocytes, we found a striking γδ T cell (>80% of thymocytes, >90% of LN cells) leukemic phenotype that, most relevant to the claim of this paper, was greatly enriched (>80%) for CD8αβ⁺ cells. These observations expand on our original report on this mouse model, where we described the expansion of CD8⁺CD4⁻ thymocytes that mostly failed to stain for TCRβ[35]. Our new data thus provide new insights into the pathogenesis of T-ALL, extending the importance of IL-7–IL-7R signaling beyond αβ T-ALL, while supporting a key role for this axis in the development of CD8αβ⁺ γδ T cells. Further, analysis of *Vav1*-hSTAT5B^N642H transgenic mice harboring the most frequent oncogenic *STAT5B* driver mutation clearly place the IL-7–IL-7R–STAT5B pathway at the core of CD8αβ⁺ γδ T cell generation. hSTAT5B^N642H-transformed γδ T cells were previously shown in transplantation experiments to reconstitute malignancy with pathological characteristics similar to those observed in participants with aggressive hepatosplenic T cell lymphoma[33]. In the current study, we found that CD8αβ⁺ γδ T cells markedly accumulated (at the expense of CD8⁻ γδ T cells) among both thymocytes and splenocytes from *Vav1*-STAT-5B^N642H transgenic mice and an altered developmental potential was confirmed in FTOCs from transgenic animals.

We believe our findings have important implications for the diagnostics and treatment of γδ T cell neoplasms, since we found a discrete

subset of participants with T-ALL expressing CD8β. This new subtype of γδ T-ALL is especially interesting given the pediatric nature of many T-ALL cases and the perinatal origin of CD8αβ⁺ γδ T cells. This is also relevant to a recent detailed characterization of γδ T cell large granular lymphocyte leukemia, a rare lymphoproliferative neoplasm characterized by the chronic proliferation of clonal large granular lymphocytes with cytotoxic activity[49], that revealed 69% of participants had a CD8⁺ γδ T cell phenotype, but identified only with a CD8α (SK1) antibody[50]. We thus propose that the characterization of all γδ T cell neoplasms should include the assessment of CD8β expression to differentially distinguish CD8αβ⁺ from CD8αα⁺ γδ T cells. We expect that future studies focused on CD8αβ⁺ γδ T cells, in both mice and humans, may be instrumental in devising new therapeutic strategies for these highly aggressive γδ T cell malignancies.

## Online content

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

## Methods

### Mice

B6 WT mice were purchased from Charles River Laboratories. PI3Kδ-deficient mice (p110δ$^{-/-}$)[51], and PI3Kδ-hyperactive mice (p110δ$^{E1020K}$)[52], were kindly provided by K. Okkenhaug (Cambridge, UK). *Rorc(γt)-Gfp$^{TG}$* reporter mice (RORγt-GFP$^{+/-}$)[21], were kindly provided by G. Eberl (Pasteur Institute, Paris). TCRα-deficient mice (TCRα$^{-/-}$)[53], TCRβ-deficient mice (TCRβ$^{-/-}$) and β2m-deficient mice (β2m$^{-/-}$) were obtained from Instituto Gulbenkian de Ciência (Oeiras, Portugal). *Rag2$^{-/-}$γc$^{-/-}$* mice were purchased (Jackson Laboratory). IL-4R$^{-/-}$ mice were kindly provided by J. Allen and P. Papotto (Manchester, UK).

Rosa26-hIL-7R.huCD2-Cre mice were previously generated and bred as described[35]. hSTAT5B$^{N642H}$ mice (official name: C57BL/6N-Tg (STAT5B < N642H >)726Biat) were previously generated as described[32]. Breeding and in vitro fertilization of these mice was approved by the institutional ethics committees of the University of Veterinary Medicine Vienna and the Champalimaud Centre for the Unknown (Lisbon, Portugal).

All strains were on a B6 background. Mice were classified as fetal (E15–17), neonatal (2–4 days), young (10–11 days), adult (4–14 weeks) or aged (31–39 weeks). Embryos were from timed pregnancies. Mice were bred and maintained in specific pathogen-free animal facilities at Queen Mary University of London or at the Instituto de Medicina Molecular João Lobo Antunes. Mouse holding conditions were as follows: 12-h light–dark cycle, temperature range 19–23 °C and humidity range 40–60%. All experiments involving animals were approved by the respective institutional ethics committees and performed in full compliance with UK Home Office and Portugal's Direção-Geral da Alimentação e Veterinária regulations and institutional guidelines. Animal experiments for the E0771 tumor challenge model were carried out under license PP0826467 to S.B.C.

### Participant material

Bone marrow or blood samples were previously collected after informed consent was obtained from participants diagnosis, according to the Declaration of Helsinki. All cases were retrospectively selected from participants enrolled in pediatric FRAALLE2000 (approved by the Leukemia's Committee of the National Scientific Committee of Société Française des Cancers de l'Enfant and by the Ethics Committee of each participating center) and adult GRAALL2005 (approved by local and multicenter research ethical committees) based on TCRγδ immunophenotype and availability of frozen diagnostic material for CD8β staining. Mononuclear cells were isolated by Ficoll density gradient before DNA extraction and cryopreservation. Immunophenotypic/molecular characterization of T-ALL samples and PDX generation were performed as previously described[54]. Immunophenotypic analysis of CD8β was performed on thawed primary samples or fresh PDX, as described below.

### *Plasmodium berghei* ANKA infection

GFP-expressing *P. berghei* ANKA sporozoites were obtained by dissection of the salivary glands of infected *Anopheles stephensi* mosquitoes bred at the IMM insectarium. Mice were injected, via the retro-orbital route, with $2 \times 10^4$ sporozoites obtained from the salivary glands of freshly dissected mosquitoes in 100 µl of DMEM medium. Noninfected mice controls received equivalent amounts of salivary glands extract from uninfected *A. stephensi* mosquitoes using the same volume and administration route. All mice were monitored daily from day 3 after infection onwards and euthanized at day 5 after infection for organ collection. Following transcardiac perfusion with ice-cold PBS, peripheral LNs, spleen and liver were collected and processed as described below.

### E0771 tumor model

B6 WT female mice (*n* = 4) aged 8 weeks were allowed to acclimatize for 2 weeks before they were injected with $0.25 \times 10^6$ E0771 murine breast adenocarcinoma cells in the mammary fat pad. Tumor growth was monitored using calipers over time, and the endpoint was considered when the tumor size reached 15 mm in diameter. Mice were euthanized by $CO_2$ asphyxiation, and tumors and distal LNs were collected for flow cytometry analysis.

### FTOCs

FTOCs were set up as previously described[12]. Briefly, fetal thymic lobes from indicated mouse strains were cultured on nucleopore membrane filter discs (Whatman) in complete RPMI 1640 medium supplemented with 10% heat-inactivated fetal calf serum (HI-FCS), 1% penicillin–streptomycin, 50 µM β-mercaptoethanol and 2 mM L-glutamine (all reagents from Invitrogen), for a specified length of time (days). In some experiments, the following antibody, inhibitors or recombinant cytokines were added to the cultures (concentrations indicated): anti-TCRδ antibody GL3 (eBioscience), MEK1/2 inhibitor UO126 (Sigma), pan-PI3K inhibitor ZSTK474 (Selleckchem), interleukin (IL)-4, IL-7 and IL-15 (concentrations indicated in figure legends, all from PeproTech). Cultures containing antibodies or inhibitors were rested overnight in fresh complete medium before analysis, unless otherwise indicated. All thymic organ cultures were subsequently analyzed by flow cytometry.

### Tissue processing and cell isolation

Tumors were processed using the mouse-specific tumor dissociation kit (Miltenyi Biotec), according to the manufacturer's instructions. Briefly, tumors were chopped into small pieces before dissociation using a heat and enzyme-assisted program on the gentleMACS dissociator (Miltenyi Biotec). Subsequently, cell suspensions were filtered through 70-µm cell strainers, red blood cells were lysed, and lymphocytes were enriched following Percoll (Sigma-Aldrich) density centrifugation. Single-cell suspensions of fetal thymocytes were obtained by gently homogenizing thymic lobes followed by straining through a 30-µm nylon gauze (Sefar) or a 40-µm cell strainer. To obtain single-cell suspensions of lymphocytes from adult mice, peripheral LNs (axillary and inguinal), thymus and spleen were dissected and strained through a 100-µm cell strainer. Livers, lungs and kidneys were dissected and cut into pieces. Small intestines were dissected, flushed with ice-cold PBS and cut open longitudinally and into pieces. Organ pieces were digested in RPMI supplemented with 10% FBS containing 1 mg ml$^{-1}$ collagenase type IV (Roche) and 100 µg ml$^{-1}$ DNase I (Sigma) for 30 min shaking at 37 °C, followed by filtering through a 100-µm cell strainer. Cells were resuspended in a 40% isotonic Percoll solution and centrifuged on an 80% Percoll solution for 20 min at 700*g* at room temperature with brake off. Leukocytes were recovered from the interface, resuspended and used for further analyses. Erythrocytes from blood, spleen, liver, lung and kidney samples were osmotically lysed in ACK lysis buffer (Invitrogen), and cells were washed in FACS buffer. Rosa26-hIL-7R. huCD2-Cre leukemic thymus and LN cells were isolated as described, homogenized in HI-FBS containing 10% dimethylsulfoxide and frozen at −80 °C until used. Samples were then thawed, washed and homogenized in complete RPMI 1640.

### Cell culture

LN-derived γδ T cells from adult WT mice were enriched by negative selection using FITC-conjugated anti-CD11b (M1/70), CD11c (HL3), CD19 (MB19-1), MHC II (M5/114) and TCRβ (H57-597) antibodies (BioLegend) and anti-FITC microbeads (Miltenyi Biotec). Labeled cells were magnetically sorted, and the flow-through with enriched γδ T cells was retained for culture under different conditions. Cells were cultured in complete RPMI medium overnight in the presence of 100 ng ml$^{-1}$ IL-12 (PeproTech) and 100 ng ml$^{-1}$ IL-18 (R&D Systems), or on plate-bound anti-CD3ε (2 µg ml$^{-1}$, clone 145-2C11, eBioscience) in the presence of soluble anti-CD28 (2 µg ml$^{-1}$, clone 37.51, eBioscience). Brefeldin A (1 µg ml$^{-1}$, eBioscience) was added for the last 5 h of culture, followed by analysis by flow cytometry. For analysis of γδ T cell subset stability,

pooled LN-derived and spleen-derived CD8αβ⁻, CD8αβ⁺ and CD8αα⁺ γδ T cells from adult WT and TCRα⁻/⁻ mice were purified by depleting CD19 (6D5)-expressing and TCRβ (H57-597)-expressing cells using biotin-labeled primary antibodies and anti-biotin MicroBeads (Miltenyi Biotec), followed by flow cytometry-assisted cell sorting of CD3⁺ (17A2) and TCRδ⁺ (GL3) γδ T cell subsets based on CD8α (53.6.7) and CD8β (YTS156.7.7) expression. Cells were labeled with CFSE following the manufacturer's instructions. Subsequently, cells were cultured in vitro in complete RPMI medium for 3 days in the presence of 20 ng ml⁻¹ IL-2, 10 ng ml⁻¹ IL-4, 40 ng ml⁻¹ IL-7, 10 ng ml⁻¹ IL-15, or solely 60 ng ml⁻¹ IL-7 (PeproTech), with or without 1 µg ml⁻¹ plate-bound anti-TCRδ (GL3). For in vivo stability experiments, purified cells were injected in the tail vein of $Rag2^{-/-}\gamma c^{-/-}$ recipient mice.

### Flow cytometry

Fluorochrome-conjugated antibodies (purchased from eBioscience, BD or BioLegend, unless otherwise indicated) against the following mouse cell surface and intracellular molecules were used: CD3 (17A2), CD3ε (145-2C11), CD8α (53.67), CD8β (H35-17.2, YTS156.7.7), CD24 (M1/69), CD27 (LG.3A10), CD44 (IM7), CD45 (30-F11), CD45RB (C363.16 A), CD73 (TY/11.8), CD122 (TM-b1), CD127 (IL-7Rα; SB/199), Eomes (W17001A), IFN-γ (XMG1.2), Ly6A/E (Sca-1; D7), IL-17A (TC11-18H10.1), Ki-67 (16A8), NK1.1 (PK136), TCRδ (GL3), TNF (MP6-XT22), Vγ1 (2.11), Vγ4 (UC3-10A6), Vγ5 (536) and Vγ7 (F2.67; kindly provided by P. Pereira, Pasteur Institute, Paris), Vδ4 (GL2) and Vδ6 (C504.17C). Fluorochrome-conjugated antibodies against human CD8β (SIDI8BEE and QA20A40) were used (purchased from Invitrogen and BioLegend, respectively). For cell surface staining, thymocytes and lymphocytes were incubated on ice with anti-CD16/CD32 (TruStain FcX; BioLegend) to block Fc receptors and stained with fluorochrome-conjugated antibodies diluted at 1:200 in FACS buffer. After staining, cells were washed and resuspended in FACS buffer. Zombie Aqua or Zombie NIR Fixable Viability dye (BioLegend) was used for dead cell exclusion. For intracellular cytokine staining, cells were stimulated (unless otherwise indicated), before staining, with 50 ng ml⁻¹ PMA (Sigma, Merck) and 1 µg ml⁻¹ ionomycin (Sigma, Merck) for 2.5 h in the presence of 10 µg ml⁻¹ brefeldin A (Merck). For intracellular detection of Eomes, IL-17A or IFN-γ cells were fixed and permeabilized with the Foxp3/Transcription Factor buffer set (eBioscience) per the manufacturer's instructions and subsequently stained with fluorochrome-conjugated antibody. Samples were acquired using FACSDiva v6.2 software (BD) on an LSR II flow cytometer (BD) or Canto II (BD), and data were analyzed using FlowJo v10.6.1 or v10.8.1 (BD).

### Single-cell RNA-seq data

Our previously generated single-cell RNA-seq data[12], publicly available on the NCBI Gene Expression Omnibus (GEO) database (accession GSE167943), were used in this study. Briefly, single-cell sequencing libraries were generated from total γδ T cells sorted from WT E15 thymic lobes cultured for 8 days in the presence or absence of pan-PI3K inhibitor ZSTK474 (0.25 µM). Analyses of RNA-seq data were carried out in R v4 using the package Seurat v5.0. Unsupervised graph-based clustering was performed using significant principal components and a clustering resolution of 0.4, and visualized using the dimensionality reduction technique, UMAP. Pseudotemporal developmental trajectories were computed using the Slingshot trajectory inference R package. Gene Ontology and enrichment pathway analysis was performed using Metascape (v3.5)[55].

### RNA-seq of participant samples

Human T-ALL RNA-seq data from St. Jude Total 15/16 cohorts ($n = 154$)[56,57], were aligned to the GRCh38 human genome reference by STAR (version 2.7.11). To quantify gene expression, RSEM (version 1.3.0) was used to calculate read counts for each transcript with the following batch correction by ComBat in the sva R package.

The DESeq2 R package was used for the normalization of each gene expression.

### Statistical analysis

No statistical methods were used to predetermine sample sizes, but our sample sizes are similar to those reported in previous publications[10–13]. Data distribution was assumed to be normal, but this was not formally tested. Two biological replicates from the neonatal small intestine analysis depicted in Fig. 2a were excluded due to absence of sufficient lymphocytes from the preparations. Data collection and analysis were not performed blind to the conditions of the experiments. All samples were processed uniformly. Statistical analysis was performed using GraphPad Prism v6.0 or v8.4.2 software. Data are presented as the mean ± s.d. Student's t-test or one-way or two-way ANOVA was used to assess statistical significance of differences between groups. A difference was considered significant if $P \leq 0.05$.

### Reporting summary

Further information on research design is available in the Nature Portfolio Reporting Summary linked to this article.

### Data availability

Single-cell RNA-seq data have been deposited in the GEO under accession GSE167943. Human T-ALL RNA-seq data are available from the European Genome-phenome Archive (EGA) under accession codes EGAS00001003975, EGAS00001004739, EGAS00001004810, EGAS00001005084, EGAS00001006336 and EGAS50000000018; the database of genotypes and phenotypes (dbGaP; http://www.ncbi.nlm.nih.gov/gap) under accession number phs002276.v2.p1 (phs000218, phs000464); the Kids First data portal (https://portal.kidsfirstdrc.org/dashboard); and the National Bioscience Database Center (NBDC) database under accession code JGAS000090. All other data that support the findings of this study are available in the article and Supplementary Information or from the corresponding authors upon reasonable request.

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

### Acknowledgements

We acknowledge the technical or logistic support of S. Martin and S. Masarone (Blizard Institute), S. Mensurado, N. Gonçalves-Sousa, V. Zuzarte-Luís and A. Parreira (Instituto de Medicina Molecular, iMM), S. Zirah (Hôpital Necker Enfants-Malades) and H. Neubauer (Institute of Animal Breeding and Genetics); and the staff of the flow cytometry

and BSU facilities at the Blizard Institute and iMM, especially G. Warnes and I. Moreira; and D. Bonaparte and the staff of the Vivarium at the Champalimaud Centre for the Unknown (Lisbon, Portugal). We also thank J. Allen and P. Papotto (University of Manchester) for the provision of IL-4R-deficient mice; and G. Turchinovich and J. Ribot for insightful discussions. This work was supported by the BBSRC (BB/R017808/1; to D.J.P.); Fundação para a Ciência e Tecnologia (PTDC/MED-ONC/6829/2020 to B.S.-S.; SFRH/BD/145352/2019 to D.I. and SFRH/BD/147411/2019 to A.C.); European Molecular Biology Organization (ALTF 252-2017 to G.J.F.), European Commission Marie Sklodowska-Curie Individual Fellowship (752932 to G.J.F.); Neue Universitätsstiftung Freiburg ('Come and STAY!' to G.J.F.); European Research Council (ERC-PoC-101069429 to J.T.B.); 'la Caixa' Foundation (HR21-00761 to J.T.B.); and Worldwide Cancer Research (WWCR 24-0426 to J.T.B.) and Breast Cancer Now (2019DecPhD1349 to S.B.C.).

## Author contributions

N.S. and G.J.F. designed and performed most of the experiments and analyzed the data. D.I., M.C.-A., A.C. and R.P. performed some experiments. R.W. and S.B.C. performed the tumor challenge experiment. S.K., L.C., V.A., L.L. and C.G.M. performed analyses of participant material. B.B., T.S. and J.D. provided technical assistance in some experiments. R.M., A.R.M.A. and S.M. provided reagents, material and support. N.S., G.J.F., J.T.B., D.J.P. and B.S.-S. conceived the study and wrote the manuscript.

## Competing interests

The authors declare no competing interests.

## Additional information

**Extended data** is available for this paper at https://doi.org/10.1038/s41590-024-01855-4.

**Correspondence and requests for materials** should be addressed to Gina J. Fiala or Daniel J. Pennington.

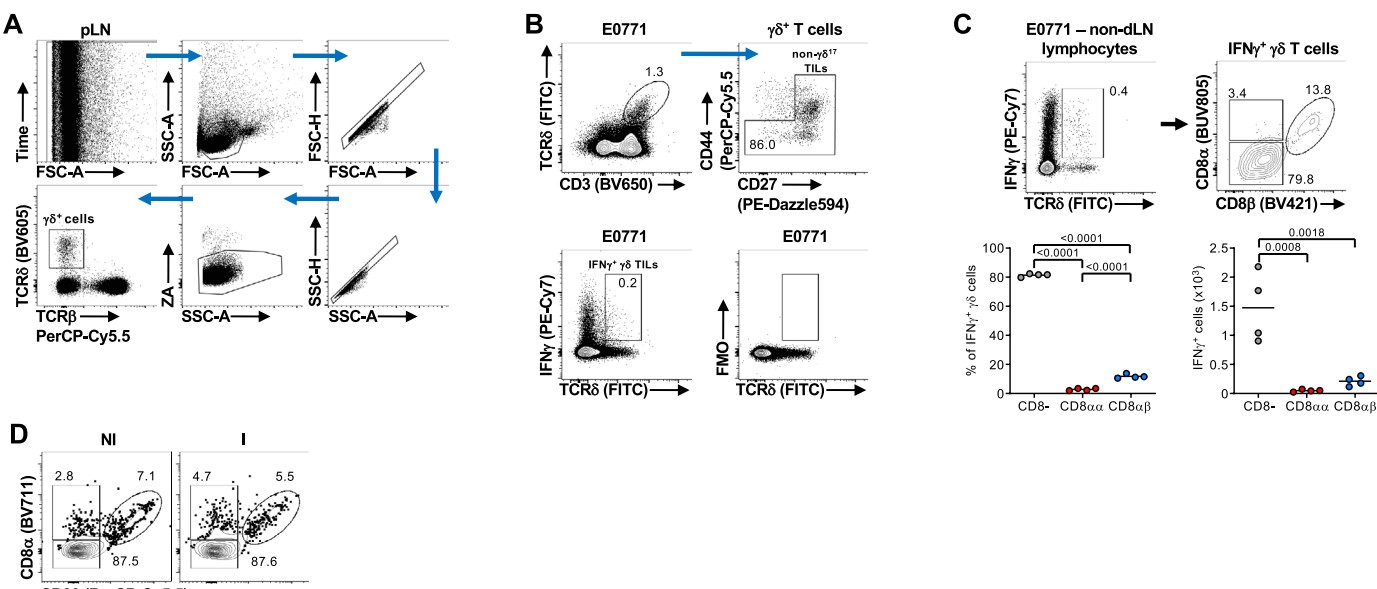

**Extended Data Fig. 1 | CD8αβ⁺ γδ T cells respond to tumour challenge and malaria infection. (a)** Flow cytometry plots represent the gating strategy used to identify mouse thymic or peripheral γδ T cells that were subsequently analysed for surface expression of CD8α and CD8β expression. The blue arrows show the chronology of the gating strategy. (**b**) Flow cytometry plots represent the gating strategy to identify non-γδ17 tumour infiltrating lymphocytes (TILs; top panel) or IFNγ⁺ γδ TILs (bottom panel) within the E0771 tumour bed. FMO, fluorescence minus one. (**c**) Lymphocytes from distal lymph nodes (LN) from E0771-bearing mice were *in vitro* stimulated with PMA/ionomycin in the presence of Brefeldin A

for subsequent intracellular IFNγ and IL-17A staining. Representative plots show gated IFNγ-expressing γδ T cells stained for CD8α/CD8β. Summary graphs show the frequency or number of CD8αα⁺, CD8αβ⁺ and CD8⁻ cells within IFNγ⁺ γδ T cells in LN. Data are from one independent experiment (*n* = 4 mice). (**d**) Lymphocytes from day 5 *Plasmodium berghei* ANKA sporozoite-infected (I) or non-infected (NI) mice were isolated from spleen and stained for within CD8α/CD8β. Representative flow cytometry plots of γδ T cells stained for CD8α/CD8β are shown (*n* = 8 mice). Percentages of gated cells are indicated in plots. Data are shown as mean ± s.d., *P* values are indicated, one-way ANOVA with Tukey's test (**C**).

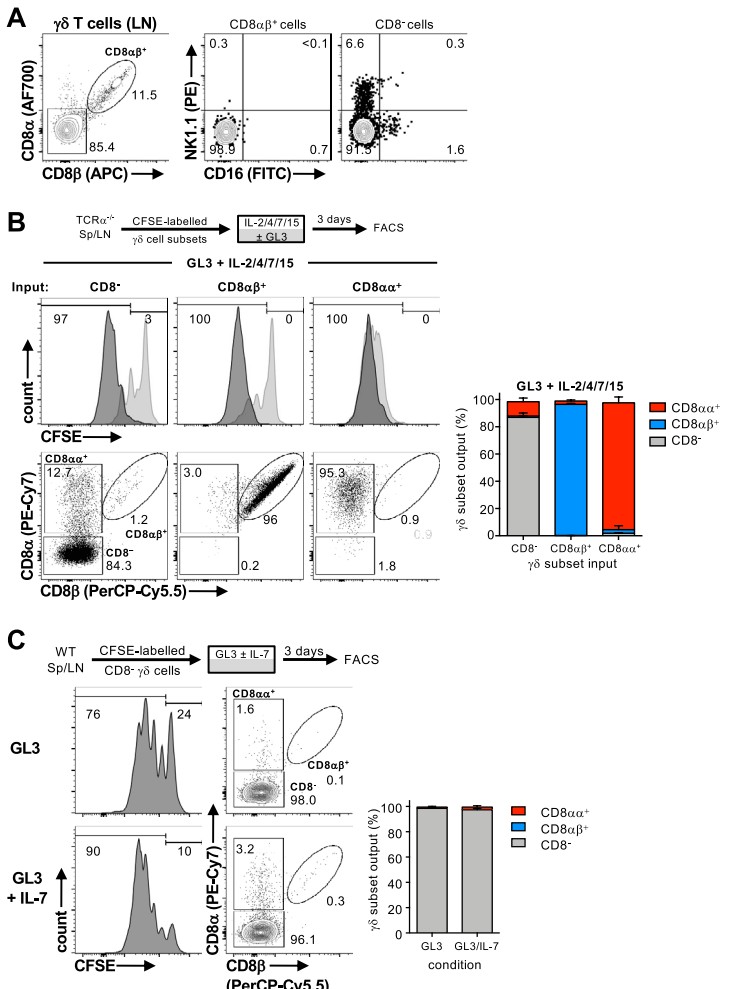

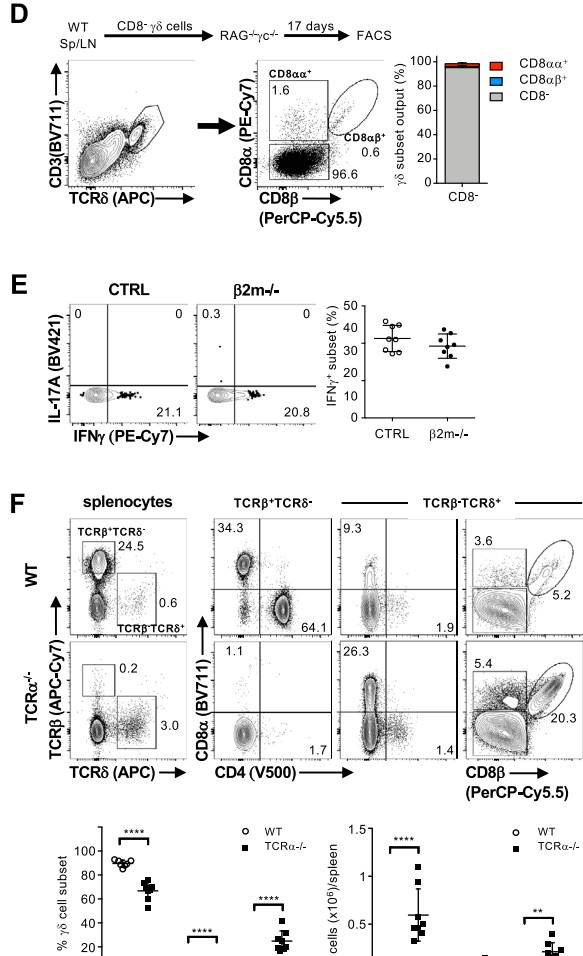

**Extended Data Fig. 2 | CD8αβ⁺ γδ T cells are a stable γδ subset.**
(**a**) Representative plots show CD8αβ⁺ and CD8⁻ γδ T cells from LNs of adult B6 WT mice stained for NK1.1/CD16. (**b**)CD8⁻, CD8αβ⁺ and CD8αα⁺ γδ T cells were purified from spleen and LNs of TCRα⁻/⁻ mice, labelled with CFSE and *in vitro* cultured in media containing a cytokine mix (IL-2/4/7/15) with (dark grey histogram) or without (light grey histogram) plate-bound GL3 (1μg/ml). After 3 days, proliferation (as by CFSE dilution), CD8α and CD8β expression were analysed by flow cytometry. Representative histograms and dot plots are shown. Summary graphs depict the composition of output cultures (*n* = 3 independent replicates pooled from 3-4 animals each). (**c**) CD8⁻ γδ T cells were purified from spleen and LNs of adult B6 WT mice, labelled with CFSE and *in vitro* cultured in the presence of plate-bound GL3 (1μg/ml) and with or without IL-7 (60 ng/ml). After 3 days, cells were analysed as in (**B**). Representative plots show CFSE dilution and CD8α and CD8β expression of output cultures. Summary graph depicts data from two independent experiments (*n* = 3 with 2–4 pooled animals each). (**d**) CD8⁻ γδ T cells were purified from spleen and LNs of B6 WT mice and injected into RAG⁻/⁻γc⁻/⁻ recipients. After 17-days, γδ T cells from recipient spleens

were analysed by flow cytometry for CD8α/CD8β. Representative dot plots are shown. Summary graph depicts data from two independent experiments (*n* = 4 mice). (**e**) Lymphocytes from LNs of 3-week old β2 m⁻/⁻ and littermate controls were *in vitro* stimulated with PMA/ionomycin in the presence of Brefeldin A. Representative plots show gated CD8αβ⁺ γδ T cells stained for intracellular IFNγ and IL-17A. Summary graphs show frequency of IFNγ⁺ CD8αβ⁺ γδ T cells with each symbol representing an individual mouse. Data are from one independent experiment (*n* = 8 mice). (**f**) Lymphocytes from B6 WT and TCRα⁻/⁻ spleens were stained for TCRβ, TCRδ, CD4, CD8α and CD8β. Representative plots show gating of αβ and γδ T cells and their subsequent analyses by CD4, CD8α and CD8β expression. Summary graphs depict the relative contribution and total numbers of CD8⁻, CD8αα⁺ and CD8αβ⁺ γδ T cells in adult mice, with each symbol representing one individual mouse. Data are representative of three independent experiments (*n* = 7-8 mice). Percentages of gated cells are indicated in plots. Data are shown as mean ± s.d., *P* values are indicated, Mann Whitney test (**E**) or two-way ANOVA with Sidak's test (**F**).

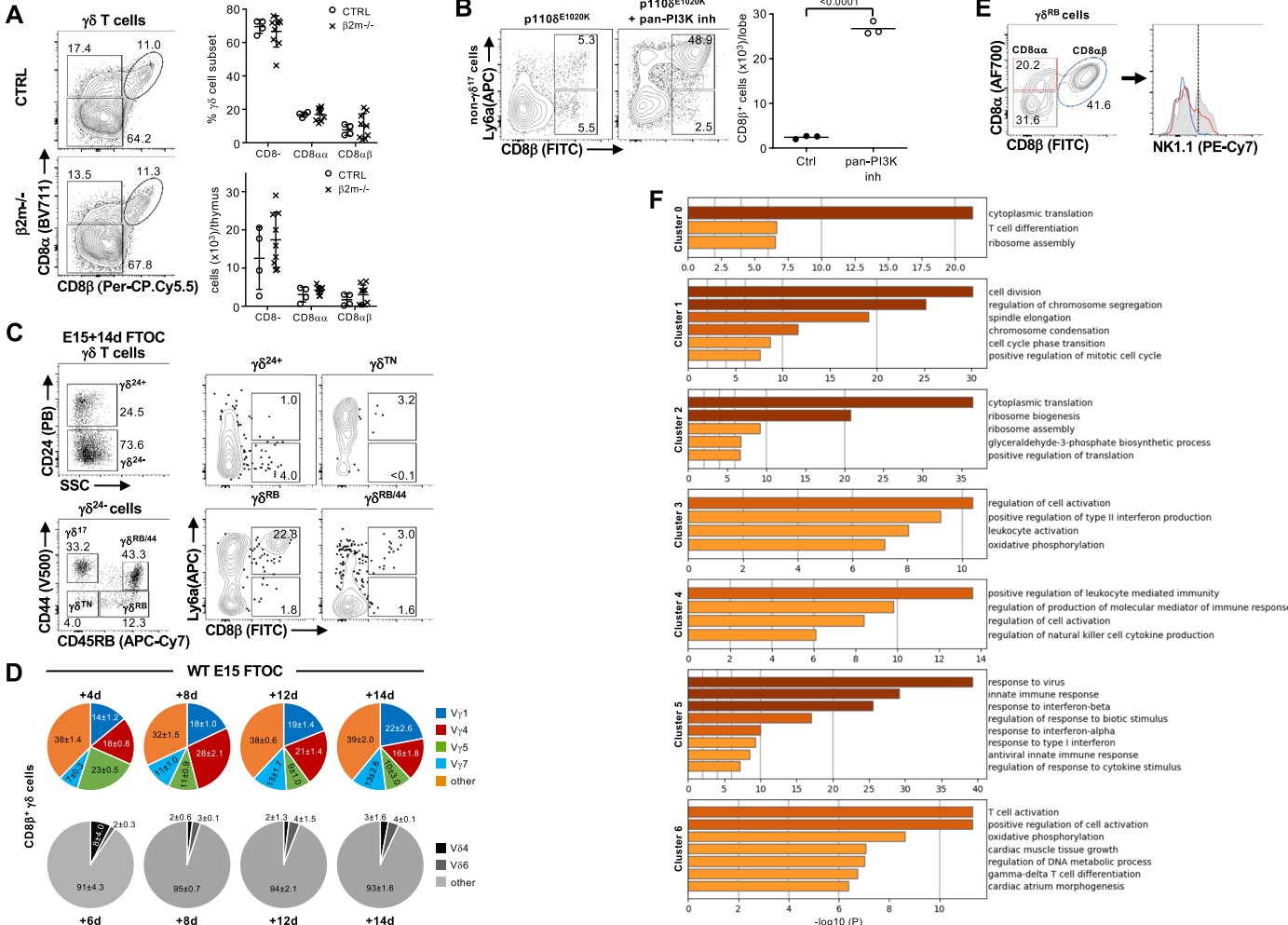

**Extended Data Fig. 3 | CD8αβ⁺ γδ T cells develop from a discrete perinatal thymic wave. (a)** Representative plots show γδ T cells from β2 m⁻/⁻ and control neonatal thymi stained for CD8α/CD8β. Data are representative of two independent experiments (*n* = 4–8 mice). Summary graphs depict the relative contribution of CD8⁻, CD8αα⁺ and CD8αβ⁺ cells to the γδ T cell compartment and total cell numbers, each symbol represents an individual mouse. **(b)** Representative flow cytometry plots show non-γδ¹⁷ T cells from p110δE1020K E15 thymic lobes cultured for 7 days in the absence or presence of the pan-PI3K inhibitor ZSTK474 (0.25 μM). Summary graph shows absolute number of CD8β⁺ γδ T cells in the cultures. Each symbol represents two thymic lobes pooled (*n* = 3 pairs). Data are representative of two independent experiments. **(c)** Representative plots of γδ T cells from WT E15 14⁻ day FTOC showing total

γδ T cells, CD24⁻ γδ (γδ²⁴⁻) T cells, and indicated γδ subsets stained for Ly6a/CD8β. Data are representative of two independent experiments. **(d)** Pie charts depict Vγ and Vδ usage by CD8αβ⁺ γδ T cells from WT E15 FTOC. Data are from one independent experiment. **(e)** Representative plot shows γδRB cells (left) with indicated subsets [CD8αα⁺ (red), CD8αβ⁺ (blue), and CD8⁻ (grey)] overlaid (right) to show NK1.1 expression from WT E15 8-day FTOC. Data are from one independent experiment. **(f)** Metascape pathway analysis of DEGs that were upregulated in the indicated clusters of integrated γδ T cells from pan-PI3K inhibitor-treated and untreated WT E15 8-day FTOC. Only the most enriched pathways are illustrated. Percentages of gated cells are indicated in plots. Data are shown as mean ± s.d., *P* values are indicated, (two-way ANOVA with Sidak's test (**A**) and unpaired two-tailed Student's *t* test (**B**).

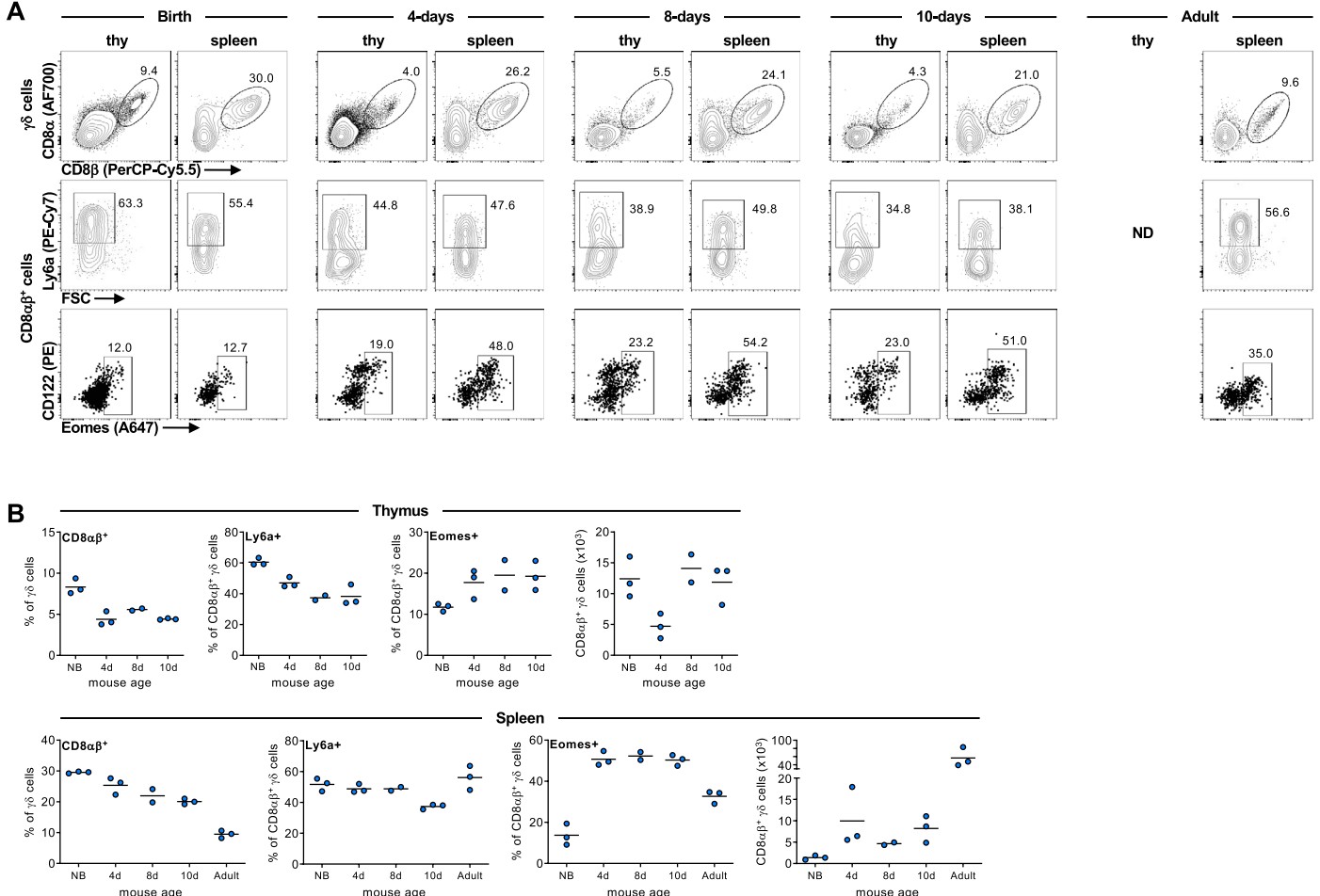

**Extended Data Fig. 4 | CD8αβ⁺ γδ T cells are maintained throughout ontogeny.** (**a**) Representative flow cytometry plots show total γδ T cells stained for CD8α/CD8β and CD8αβ⁺ γδ T cells stained for Ly6a or CD122/Eomes from thymus and spleen of WT mice during ontogeny. (**b**) Summary graphs show the frequency of CD8αβ⁺ γδ T cells, Ly6a⁺, and Eomes⁺ CD8αβ⁺ cells, and absolute cell number of CD8αβ⁺ γδ T cells. Data are from one experiment (*n* = 3 mice per time-point except 8-days, *n* = 2 mice). Percentages of gated cells are indicated in plots.

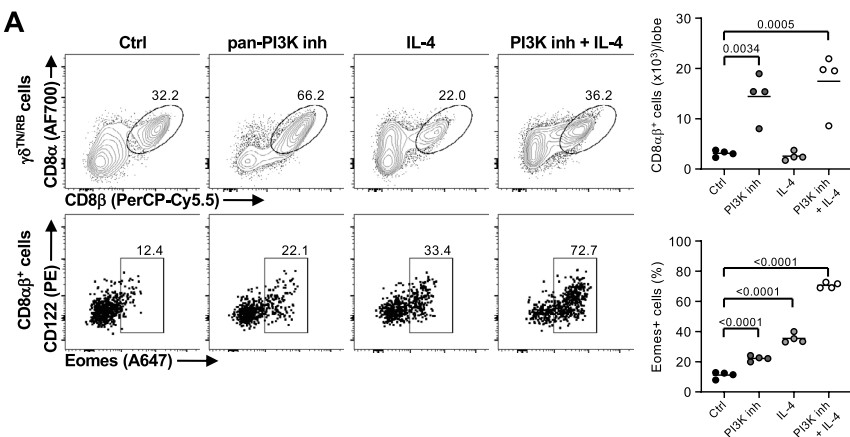

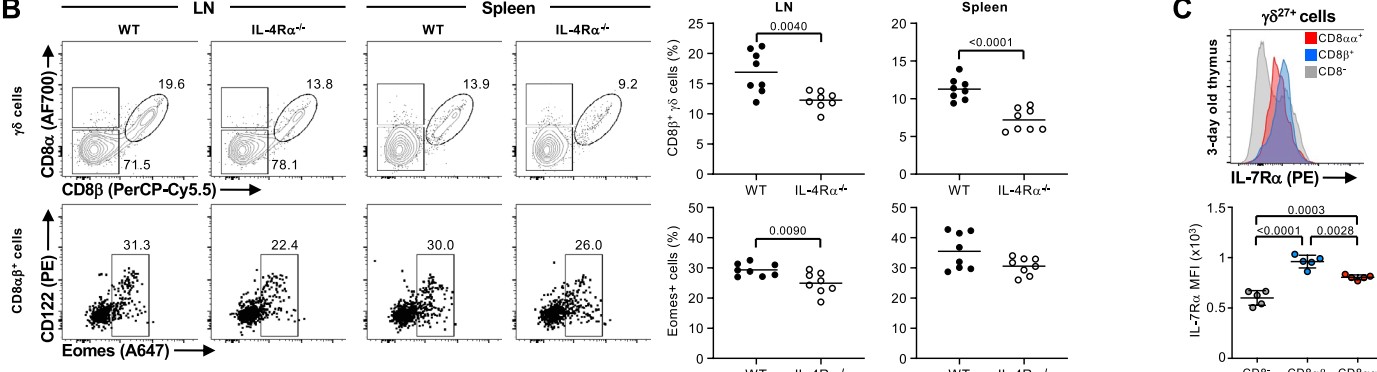

**Extended Data Fig. 5 | IL-4, IL-7 and low TCR signaling promote CD8αβ⁺ γδ T cell development.** (a) Representative plots show γδ^TN/RB cells (top panel) and CD8αβ⁺ γδ^TN/RB cells stained for CD122/Eomes (bottom panels) from 8-day FTOC of WT E15 thymic lobes in the presence or absence of pan-PI3K inhibitor (ZSTK474; 0.25 μM) and/or IL-4 (20 ng/ml). Summary graphs show the absolute numbers of CD8αβ⁺ cells (top), or the percentage of Eomes⁺ CD8αβ⁺ cells (bottom) in the cultures (n = 4 samples/group). Each symbol in summary graphs represents at least three lobes pooled. Data are representative of two independent experiments. (b) Representative plots show γδ T cells (top panel) and CD8αβ⁺ γδ T cells stained for CD122/Eomes (bottom panels) from lymph

nodes and spleen of WT and IL-4Rα^-/- mice. Summary graphs show the frequency of CD8αβ⁺ γδ T cells and Eomes⁺ CD8αβ⁺ cells. Each symbol in summary graphs represents an individual mouse. Data are pooled from two independent experiments (n = 8 mice). (c) Representative histograms show expression of IL-7Rα in the indicated subsets [CD8αα⁺ (red), CD8αβ⁺ (blue), and CD8⁻ (grey)] within γδ^27+ cells from B6 WT neonates. Summary graph shows MFI of IL-7Rα in the indicated subsets with each symbol representing an individual mouse. Data are representative of one experiment (n = 5). Percentages of gated cells are indicated. Data are shown as mean ± s.d. P are indicated, one-way ANOVA with Dunnett's Tukey's test (**A** and **C**) and unpaired two-tailed Student's t test (**B**).

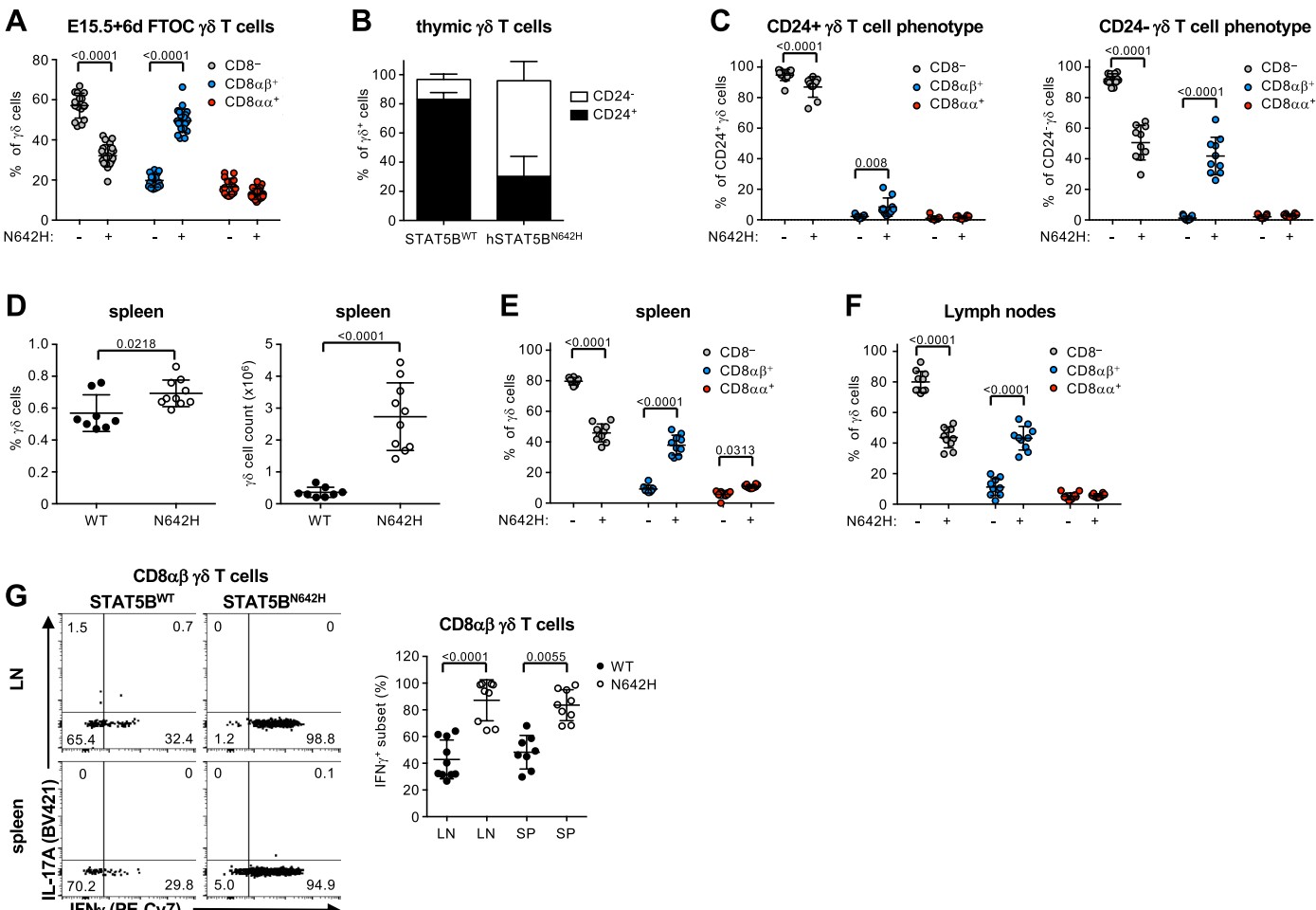

**Extended Data Fig. 6 | Oncogenic hSTAT5B promotes the expansion of CD8αβ⁺ γδ T cells.** (a) Summary graph shows percentages of CD8⁻, CD8αβ⁺ and CD8αα⁺ γδ T cells in *Vav1*.hSTAT5B^N642H (N642H+) and STAT5B^WT (N642H-) littermate control E15.5 + 6d FTOC. Data are from two independent experiments and each symbol represents one lobe (*n* = 16–22 lobes). (b) Summary graph depicts percentages of CD24⁺ and CD24⁻ γδ T cells in N642H and WT littermate control adult thymus. Data are from two independent experiments and each symbol represents an individual mouse (*n* = 10–13 mice). (c) Summary graphs depict percentages of CD8⁻, CD8αβ⁺ and CD8αα⁺ within CD24⁺ or CD24⁻ γδ T cells in adult thymus from N642H and WT littermate controls. Data are from two independent experiments and each symbol represents an individual mouse (*n* = 10–13 mice). (d) Summary graphs depict percentages and total cell counts of γδ T cells in N642H and WT littermate control adult spleen. Data are from two independent experiments and each symbol represents an individual mouse

(*n* = 8–9 mice). (e and f) Summary graph depicts percentages of CD8⁻, CD8αβ⁺ and CD8αα⁺ γδ T cells in N642H and WT littermate control adult spleen (**E**) and LNs (**F**). Data are from two independent experiments and each symbol represents an individual mouse (*n* = 8–10 mice). (g) Lymphocytes from LNs and spleen of N642H and WT littermate controls were *in vitro* stimulated with PMA/ionomycin in the presence of Brefeldin A. Representative plots show gated CD8αβ⁺ γδ T cells stained for intracellular IFNγ and IL-17A. Summary graph depicts percentages of IFNγ⁺ CD8αβ⁺ γδ T cells in LNs and spleen from N642H and WT littermate controls. Data are from two independent experiment and each symbol represents one invidual mouse (*n* = 8–10 mice). Percentages of gated cells are indicated in plots. Data are shown as mean ± s.d. *P* values are indicated, two-way ANOVA with Sidak's or Dunn's test (**A**, **C**, **E** - **G**), Mann Whitney test (**D**) and unpaired Student's *t* test (**D**).

**Extended Data Table 1 | Clinical details of the three CD8B<sup>hi</sup> γδ T-ALL participants in Fig. 6a**

| | Patient 1 | Patient 2 | Patient 3 |
|---|---|---|---|
| **Genomic subtype** | HOXA (DDX3X::MLLT10) | HOXA (KMT2A::AFDN) | Immature |
| **Age** | 1 - 5 | 11 - 15 | 6 - 10 |
| **Sex** | Female | Male | Male |
| **TCRD** | V1/D3/J1/C | V1/D3/J1/C | V1/D3/J1/C |
| **TCRD** | CALGESTGGTDKLIF | CALGNSLAAKPNTGGYPTDKLIF | CALGELNPSFKLGDMGLIF |
| **TCRG** | V9/J1-J2/C2-C1 | V9/JP | V8/JP2/C2-C1 |
| **TCRG** | CALWEDNYKKLF | CALWEVRELGKKIKVF | CATWDMGSDWIKTF |

# Reporting Summary

## Statistics

For all statistical analyses, confirm that the following items are present in the figure legend, table legend, main text, or Methods section.

| n/a | Confirmed | |
|---|---|---|
| ☐ | ☒ | The exact sample size (*n*) for each experimental group/condition, given as a discrete number and unit of measurement |
| ☐ | ☒ | A statement on whether measurements were taken from distinct samples or whether the same sample was measured repeatedly |
| ☐ | ☒ | The statistical test(s) used AND whether they are one- or two-sided *Only common tests should be described solely by name; describe more complex techniques in the Methods section.* |
| ☐ | ☒ | A description of all covariates tested |
| ☒ | ☐ | A description of any assumptions or corrections, such as tests of normality and adjustment for multiple comparisons |
| ☐ | ☒ | A full description of the statistical parameters including central tendency (e.g. means) or other basic estimates (e.g. regression coefficient) AND variation (e.g. standard deviation) or associated estimates of uncertainty (e.g. confidence intervals) |
| ☐ | ☒ | For null hypothesis testing, the test statistic (e.g. *F*, *t*, *r*) with confidence intervals, effect sizes, degrees of freedom and *P* value noted *Give P values as exact values whenever suitable.* |
| ☒ | ☐ | For Bayesian analysis, information on the choice of priors and Markov chain Monte Carlo settings |
| ☒ | ☐ | For hierarchical and complex designs, identification of the appropriate level for tests and full reporting of outcomes |
| ☒ | ☐ | Estimates of effect sizes (e.g. Cohen's *d*, Pearson's *r*), indicating how they were calculated |

*Our web collection on statistics for biologists contains articles on many of the points above.*

## Software and code

Policy information about availability of computer code

| Data collection | Flow cytometry data was acquired using FACSDiva v6.2 software (BD Bioscience). |
|---|---|
| Data analysis | All analysis are described in the relevant section of Methods. Flow cytometry data was analysed using FlowJo software v10.6.1 or v10.8.1 Single-cell RNAseq data was analysed using the R package Seurat v5.0 and the Slingshot trajectory inference R package Gene ontology and enrichment pathway analysis was performed using Metascape v3.5 human RNA-Seq data were aligned to the GRCh38 human genome reference by STAR1015 (version 2.7.11) and analysed using RSEM (version 1.3.0) with the batch correction by ComBat in the sva R1017 package; DESeq2 R package was used for the normalization of each gene expression. Statistical analyses were done with GraphPad Prism v6.0 or v8.4.2 |

For manuscripts utilizing custom algorithms or software that are central to the research but not yet described in published literature, software must be made available to editors and reviewers. We strongly encourage code deposition in a community repository (e.g. GitHub). See the Nature Portfolio guidelines for submitting code & software for further information.

# Data

Policy information about availability of data

All manuscripts must include a data availability statement. This statement should provide the following information, where applicable:
- Accession codes, unique identifiers, or web links for publicly available datasets
- A description of any restrictions on data availability
- For clinical datasets or third party data, please ensure that the statement adheres to our policy

> The accession code for single cell RNA sequencing data is GSE167943. The data that support the findings of this study are available from the corresponding authors upon request.

# Research involving human participants, their data, or biological material

Policy information about studies with human participants or human data. See also policy information about sex, gender (identity/presentation), and sexual orientation and race, ethnicity and racism.

| | |
|---|---|
| Reporting on sex and gender | Sex and gender were not taken into account in the study design for human samples. Given the small number of gamma delta T-ALL patient samples that we could have access to, we used all that were available for this study. Nonetheless, we found that there was a male/female ratio of 2.08, which is within the known range reported in the literature for T-ALL. |
| Reporting on race, ethnicity, or other socially relevant groupings | We did not consider race, ethnicity or other socially relevant grouping in our analyses. |
| Population characteristics | Median age : 25 years; Age range : 4-70 years; Sex ratio (M/F): 2.08; Diagnosis: T-ALL. Samples were collected at diagnosis, before treatment. We used 37 gamma-delta primary T-ALL samples and 5 PDX samples derived from primary T-ALL samples, previously collected at Necker Enfants-Malades Hospital (Paris, France). Mononuclear cells were isolated by ficoll density gradient and cryopreserved in liquid nitrogen. |
| Recruitment | We used cryopreserved samples from patients that were gamma-delta T-ALL, available at Necker Enfants-Malades Hospital (N=37+5). |
| Ethics oversight | All adult and pediatric cases were retrospectively selected from FRAALLE2000 and GRAALL2005 enrolled patients based on their TCRgd expressing immunophenotype and availability of frozen diagnostic material for CD8b staining. Pediatric cases were treated according to the FRALLE2000 T guidelines. Informed consent for data registration was provided according to the Declaration of Helsinki. This study was approved by the Leukemia's Committee of the National Scientific Committee of the SFCE (Société Française des Cancers de l'Enfant) and by the Ethics Committee of each participating center. Adults cases were enrolled within the multicenter randomized GRAAL-2005 protocol. Informed consent was obtained from all patients at trial entry. This study was conducted in accordance with the Declaration of Helsinki and approved by local and multicenter research ethical committees. |

Note that full information on the approval of the study protocol must also be provided in the manuscript.

# Field-specific reporting

Please select the one below that is the best fit for your research. If you are not sure, read the appropriate sections before making your selection.

☒ Life sciences   ☐ Behavioural & social sciences   ☐ Ecological, evolutionary & environmental sciences

For a reference copy of the document with all sections, see nature.com/documents/nr-reporting-summary-flat.pdf

# Life sciences study design

All studies must disclose on these points even when the disclosure is negative.

| | |
|---|---|
| Sample size | In a single experiment, at least three biological replicates were used. No sample size calculations were performed, but the sample sizes were determined based on our experimental observations and our experiences in giving reliable and reproducible results. Furthermore, data was collected in repeated independent experiments. |
| Data exclusions | Two biological replicates from neonatal small intestine analysis depicted in Fig. 2A were excluded due to absence of sufficient lymphocytes from the preparations. |
| Replication | At least three biological replicates were used for all experiments, except for one experiment of neonatal organs with two biological replicates pooled from 4 neonates each. Experimental results depicted in Fig. 1B-C and Ext. Data Fig. 1B-C represent data from one experiment with four independent biological replicates. Experimental results depicted in Fig. 1E-F, Ext. Data Fig. 1C, Fig. 2B, D, E, Ext. Data Fig.2C-D, Ext. Data Fig. 2F, Fig. 4D-E, Fig. 4G, Ext. Data Fig. 5B, Fig. 5A-F, and in Ext. Data Fig. 6A-G represent merged data from at least two independent experiments each. |

Experimental results depicted in Fig. 2A summarizes eight (neonatal spleen), seven (neonatal LN), six (neonatal liver and adult spleen), five (adult LN), four (adult liver), three (neonatal lung) two (neonatal and adult intestine, neonatal kidney) and one experiments (adult lung and kidney).

Experimental results depicted in Fig. 2C, Ext. Data Fig. 2A, Fig. 3B-D, Ext. Data Fig. 3A, B, C, E, Fig. 4A-C, Fig. 4F, and Ext. Data Fig. 5A were independently repeated two times with similar results.

Experimental results in Ext. Data Fig. 2A depicts data from one experiment with three independent biological replicates originating from three to four pooled independent donors each. Experimental results in Ext. Data Fig. 2D, Ext. Data Fig. 3C, Ext. Data Fig. 4, and Ext. Data Fig. 5C depict data from one experiment each.

Experimental results depicted in Fig. 3A summarizes five experiments (neonatal and adult), three (young thymus) and two (embryonic thymus and old thymus), respectively.

Experimental results in Fig. 6 depict analyses of two patient cohort with 154 or 42 patients each.

Representative plots and graphs are presented and representative or pooled summaries shown. Descriptions of the number of replicates and independent experiments are included in the corresponding figure legends as well as information on the statistical tests performed and calculated P values.

| | |
|---|---|
| Randomization | Randomization is not relevant to this study as samples in each experiment were treated uniformly and the same data analysis procedure was applied to all samples of the same experiment. |
| Blinding | Investigators were not blinded in this study because all results presented are based on quantitative analysis which is not subject to human biases. |

# Reporting for specific materials, systems and methods

We require information from authors about some types of materials, experimental systems and methods used in many studies. Here, indicate whether each material, system or method listed is relevant to your study. If you are not sure if a list item applies to your research, read the appropriate section before selecting a response.

## Materials & experimental systems

| n/a | Involved in the study |
|---|---|
| ☐ | ☒ Antibodies |
| ☒ | ☐ Eukaryotic cell lines |
| ☒ | ☐ Palaeontology and archaeology |
| ☐ | ☒ Animals and other organisms |
| ☒ | ☐ Clinical data |
| ☒ | ☐ Dual use research of concern |
| ☒ | ☐ Plants |

## Methods

| n/a | Involved in the study |
|---|---|
| ☒ | ☐ ChIP-seq |
| ☐ | ☒ Flow cytometry |
| ☒ | ☐ MRI-based neuroimaging |

## Antibodies

| | |
|---|---|
| Antibodies used | Antibody (clone), Fluorochrome Catalogue # Supplier<br><br>CD3 (17A2), BV650, 100229 Biolegend<br>CD3 (17A2), BV711, 100241 Biolegend<br>CD3e (145-2C11), unconjugated, 16-0031-85 eBioscience<br>CD4 (RM4-5), V500, 560782 BD<br>CD5 (53-7.3), PE, 553023 BD<br>CD8a (53-6.7), BUV805, 612898 BD<br>CD8a (53-6.7), BV711, 100759 Biolegend<br>CD8a (53-6.7), PE-Cy7, 100721 Biolegend<br>CD8a (53-6.7), PE, 100708 Biolegend<br>CD8a (53-6.7), AF700, 100730 Biolegend<br>CD8b (YTS156.7.7), BV421, 126629 Biolegend<br>CD8b (YTS156.7.7), FITC, 126606 Biolegend<br>CD8b (YTS156.7.7), APC, 126614 Biolegend<br>CD8b (H35-17.2), PerCP-Cy5.5, 46-0083-82 eBioscience<br>CD8b (SIDI8BEE), PE, 12-5273-42 eBioscience<br>CD8b (QA20A40), PE, 376703 Biolegend<br>CD11b (M1/70), FITC, 101205 Biolegend<br>CD11c (N418), FITC, 117306 Biolegend<br>CD16 (S17014E), FITC, 158007 Biolegend<br>CD19 (MB19-1), FITC, 101506 Biolegend<br>CD19 (6D5), biotin, 115504 Biolegend<br>CD24 (M1/69), Pacific Blue, 101820 Biolegend<br>CD27 (LG.3A10), PE-Dazzle594, 124228 Biolegend<br>CD28 (37.51), unconjugated, 16-0281-85 eBioscience<br>CD44 (IM7), PerCP-Cy5.5, 103032 Biolegend<br>CD44 (IM7), V500, 560780 BD<br>CD45 (30-F11), BV510, 103138 Biolegend<br>CD45RB (C363.16A), APC-Cy7, 103310 Biolegend |

CD73 (TY/11.8), PE-Cy7, 25-0731-80 eBioscience
CD122 (TM-b1), PE, 12-1222-82 eBioscience
CD127 (IL-7Ra; SB/199), PE, 12-1273-82 eBioscience
Eomes (W17001A), A647, 157703 Biolegend
IFNg (XMG1.2), PE-Cy7, 25-7311-82 eBioscience
IL-17A (TC11-18H10.1), BV421, 506926 Biolegend
Ki67 (16A8), PE, 652404 Biolegend
Ly6A/E (D7), BV605, 108133 Biolegend
Ly6A/E (D7), APC, 108112 Biolegend
Ly6A/E (D7), PE-Cy7, 108114 Biolegend
MHCII (M5/114), FITC, 107606 Biolegend
NK1.1 (PK136), PE, 108708 Biolegend
NK1.1 (PK136), PE-Cy7, 108714 Biolegend
TCRb (H57-597), PerCP-Cy5.5, 109228 Biolegend
TCRb (H57-597), APC-Cy7, 109220 Biolegend
TCRb (H57-597), FITC, 109206 Biolegend
TCRb (H57-597), biotin, 109204 Biolegend
TCRd (GL3), FITC, 11-5711-85 eBioscience
TCRd (GL3), APC, 17-5711-82 eBioscience
TCRd (GL3), BV605, 118129 Biolegend
TNFa (MP6-XT22), APC, 506308 Biolegend
Vg1 (2.11), PE, 141106 Biolegend
Vg4 (UC3-10A6), FITC, 137704 Biolegend
Vg5 (536), PE, 137504 Biolegend
Vg7 (F2.67), FITC, kindly provided by Dr Pablo Pereira, Pasteur Institute, Paris, France
Vd4 (GL2), PE, 134905 Biolegend
Vd6 (C504.17C), FITC, 154807 Biolegend

Validation

The antibodies used in this study were used according to the manufacturer's recommendation. Validation was based on the description provided on the manufacturers' homepage.
Antibody (clone) Validation
CD3 (17A2) https://www.biolegend.com/en-gb/products/brilliant-violet-650-anti-mouse-cd3-antibody-7843?GroupID=BLG242
https://www.biolegend.com/en-gb/products/brilliant-violet-711-anti-mouse-cd3-antibody-10022
CD3e (145-2C11) https://www.thermofisher.com/antibody/product/CD3e-Antibody-clone-145-2C11-Monoclonal/14-0031-82
CD4 (RM4-5) https://www.bdbiosciences.com/en-us/products/reagents/flow-cytometry-reagents/research-reagents/single-color-antibodies-ruo/v500-rat-anti-mouse-cd4.560782
CD5 (53-7.3) https://www.bdbiosciences.com/en-us/products/reagents/flow-cytometry-reagents/research-reagents/single-color-antibodies-ruo/pe-rat-anti-mouse-cd5.553023
CD8a (53-6.7) https://www.bdbiosciences.com/en-gb/products/reagents/flow-cytometry-reagents/research-reagents/single-color-antibodies-ruo/buv805-rat-anti-mouse-cd8a.612898
https://www.biolegend.com/en-gb/products/alexa-fluor-700-anti-mouse-cd8a-antibody-3387
https://www.biolegend.com/de-de/products/brilliant-violet-711-anti-mouse-cd8a-antibody-7926?GroupID=BLG279
https://www.biolegend.com/en-gb/products/pe-cyanine7-anti-mouse-cd8a-antibody-1906?GroupID=BLG2559
https://www.biolegend.com/en-gb/products/pe-anti-mouse-cd8a-antibody-155?GroupID=BLG2559
CD8b (YTS156.7.7) https://www.biolegend.com/fr-lu/products/brilliant-violet-421-anti-mouse-cd8b-ly-3-antibody-17374
https://www.biolegend.com/en-gb/products/fitc-anti-mouse-cd8b-antibody-4475?GroupID=BLG4212
https://www.biolegend.com/en-gb/products/apc-anti-mouse-cd8b-antibody-9055
(H35-17.2) https://www.thermofisher.com/antibody/product/CD8b-Antibody-clone-eBioH35-17-2-H35-17-2-Monoclonal/46-0083-82
CD11b (M1/70) https://www.biolegend.com/en-gb/products/fitc-anti-mouse-human-cd11b-antibody-347
CD11c (N418) https://www.biolegend.com/en-gb/products/fitc-anti-mouse-cd11c-antibody-1815?GroupID=BLG11937
CD16 (S17014E) https://www.biolegend.com/en-gb/sean-tuckers-tests/fitc-anti-mouse-cd16-antibody-19303?GroupID=ImportedGROUP1
CD19 (MB19-1) https://www.biolegend.com/en-gb/sean-tuckers-tests/fitc-anti-mouse-cd19-antibody-1971?GroupID=BLG4752
CD19 (6D5) https://www.biolegend.com/en-ie/products/biotin-anti-mouse-cd19-antibody-1527?GroupID=BLG7045
CD24 (M1/69) https://www.biolegend.com/en-gb/products/pacific-blue-anti-mouse-cd24-antibody-3584
CD27 (LG.3A10) https://www.biolegend.com/en-ie/products/pe-dazzle-594-anti-mouse-rat-human-cd27-antibody-11906
CD28 (37.51) https://www.thermofisher.com/antibody/product/CD28-Antibody-clone-37-51-Monoclonal/16-0281-82
CD44 (IM7) https://www.bdbiosciences.com/en-nz/products/reagents/flow-cytometry-reagents/research-reagents/single-color-antibodies-ruo/v500-rat-anti-mouse-cd44-pgp-1-ly-24.560780
CD45 (30-F11) https://www.biolegend.com/ja-jp/products/brilliant-violet-510-anti-mouse-cd45-antibody-7995?GroupID=BLG1932
CD45RB (C363.16A) https://www.biolegend.com/en-gb/products/apc-cyanine7-anti-mouse-cd45rb-antibody-3525?GroupID=BLG259
CD73 (TY/11.8) https://www.thermofisher.com/antibody/product/CD73-Antibody-clone-eBioTY-11-8-TY-11-8-Monoclonal/25-0731-82
CD122 (TM-b1) https://www.thermofisher.com/antibody/product/CD122-Antibody-clone-TM-b1-TM-beta1-Monoclonal/12-1222-82
CD127 (SB/199) https://www.thermofisher.com/antibody/product/CD127-Antibody-clone-eBioSB-199-SB-199-Monoclonal/12-1273-82
Eomes (W17001A) https://www.biolegend.com/en-gb/products/alexa-fluor-647-anti-mouse-eomes-antibody-18078
IFNg (XMG1.2) https://www.thermofisher.com/antibody/product/IFN-gamma-Antibody-clone-XMG1-2-Monoclonal/25-7311-82
IL-17A (TC11-18H10.1) https://www.biolegend.com/en-gb/products/brilliant-violet-421-anti-mouse-il-17a-antibody-7223
Ki67 (16A8) https://www.biolegend.com/en-gb/products/pe-anti-mouse-ki-67-antibody-8134?GroupID=GROUP26
Ly6A/E (D7) https://www.biolegend.com/en-gb/products/brilliant-violet-605-anti-mouse-ly-6a-e-sca-1-antibody-8664
https://www.biolegend.com/en-gb/products/apc-anti-mouse-ly-6a-e-sca-1-antibody-225
https://www.biolegend.com/en-gb/products/pe-cyanine7-anti-mouse-ly-6a-e-sca-1-antibody-3137
MHCII (M5/114) https://www.biolegend.com/en-gb/products/fitc-anti-mouse-i-a-i-e-antibody-366
NK1.1 (PK136) https://www.biolegend.com/en-gb/products/pe-anti-mouse-nk-1-1-antibody-431
https://www.biolegend.com/en-gb/products/pe-cyanine7-anti-mouse-nk-1-1-antibody-2840

TCRb(H57-597) https://www.biolegend.com/en-gb/products/percp-cyanine5-5-anti-mouse-tcr-beta-chain-antibody-5603
https://www.biolegend.com/en-ie/products/apc-cyanine7-anti-mouse-tcr-beta-chain-antibody-4137?GroupID=BLG6994
https://www.biolegend.com/en-gb/products/fitc-anti-mouse-tcr-beta-chain-antibody-270
https://www.biolegend.com/en-ie/products/biotin-anti-mouse-tcr-beta-chain-antibody-269
TCRd (GL3) https://www.thermofisher.com/antibody/product/TCR-gamma-delta-Antibody-clone-eBioGL3-GL-3-GL3-Monoclonal/11-5711-85
https://www.thermofisher.com/antibody/product/TCR-gamma-delta-Antibody-clone-eBioGL3-GL-3-GL3-Monoclonal/17-5711-82
https://www.biolegend.com/en-gb/products/brilliant-violet-605-anti-mouse-tcr-gamma-delta-antibody-9655
TNFa (MP6-XT22) https://www.biolegend.com/en-gb/products/apc-anti-mouse-tnf-alpha-antibody-975?GroupID=GROUP24
Vg1 (2.11) https://www.biolegend.com/en-gb/products/pe-anti-mouse-tcr-vgamma1-1-cr4-antibody-7039
Vg4 (UC3-10A6) https://www.biolegend.com/en-gb/products/fitc-anti-mouse-tcr-vgamma2-antibody-6536
Vg5 (536) https://www.biolegend.com/en-gb/products/pe-anti-mouse-tcr-vgamma3-antibody-6525
Vg7 (F2.67) https://www.biolegend.com/en-gb/products/purified-anti-mouse-tcr-vgamma7-antibody-20318
Vd4 (GL2) https://www.biolegend.com/en-gb/products/pe-anti-mouse-tcr-vdelta4-antibody-6105
Vd6 (C504.17C) https://www.biolegend.com/en-gb/products/fitc-anti-mouse-tcr-vdelta6-3-antibody-16294

# Animals and other research organisms

Policy information about studies involving animals; ARRIVE guidelines recommended for reporting animal research, and Sex and Gender in Research

| Laboratory animals | C57BL/6 wild-type (B6 WT), PI3Kδ-deficient mice (p110d-/-), PI3Kδ-hyperactive mice (p110dE1020K), Rorc(gt)-GfpTG reporter mice (RORgt-GFP+/-), TCRa-deficient mice (TCRa-/-), TCRb-deficient mice (TCRb-/-), and b2m-deficient mice (b2m-/-), Rag2-/-gc-/- mice, IL-4R-/- mice, Rosa26-hIL-7R.huCD2-Cre mice, hSTAT5BN642H mice (official name: C57BL/6N-Tg(STAT5B<N642H>)726Biat). All strains were on a C57BL/6 background. Mice were foetal (E15-E17), neonatal (2-4 days), young (10-11 days), adult (4-14 weeks) or aged (31-39 weeks). Embryos were from timed pregnancies or in vitro fertilization. Both male and female animals were used in this study and matched with controls of corresponding age and sex. |
| --- | --- |
| Wild animals | The study does not involve wild animals. |
| Reporting on sex | Results presented include both female and male animals. Sex was determined phenotypically whenever possible. No information on embryo sex was collected. For young and old animals sex of individuals was determined but no sex-specific effects were observed in the tissues analysed. In a limited set of neonates sex was determined but no sex-specific effects observed in the tissues analysed (thymus, spleen LN and liver). In adult mice, there might be a tendency of increased percentages of CD8ab gd T cells in male thymus, LN and spleen compared to females. A detailed analysis has not been performed. Data on other tissues (lungs, kidneys and small intestine) originates from female animals only. |
| Field-collected samples | The study did not involve field-collected samples. |
| Ethics oversight | All experiments involving animals were approved by the respective institutional ethics committees and performed in full compliance with UK Home Office and Portugal's Direção-Geral da Alimentação e Veterinária regulations and institutional guidelines. Breeding and in vitro fertilization of C57BL/6N-Tg(STAT5B<N642H>)726Biat mice was approved by the institutional ethics committees of University of Veterinary Medicine Vienna and the Champalimaud Centre for the Unknown (Lisbon, Portugal). |

Note that full information on the approval of the study protocol must also be provided in the manuscript.

# Plants

| Seed stocks | n.a. |
| --- | --- |
| Novel plant genotypes | n.a. |
| Authentication | n.a. |

# Flow Cytometry

## Plots

Confirm that:

☒ The axis labels state the marker and fluorochrome used (e.g. CD4-FITC).

☒ The axis scales are clearly visible. Include numbers along axes only for bottom left plot of group (a 'group' is an analysis of identical markers).

☒ All plots are contour plots with outliers or pseudocolor plots.

☒ A numerical value for number of cells or percentage (with statistics) is provided.

## Methodology

| | |
|---|---|
| Sample preparation | Tumours were processed using the mouse-specific tumour dissociation kit (Miltenyi Biotec), according to the manufacturer's instructions. Briefly, tumours were chopped into small pieces before dissociation using a heat and enzyme-assisted program on the gentleMACS dissociator (Miltenyi Biotec). Subsequently, cell suspensions were filtered through 70 um cell strainers, red blood cells were lysed, and lymphocytes were enriched following Percoll (Sigma-Aldrich) density centrifugation. Single-cell suspensions of foetal thymocytes were obtained by gently homogenizing thymic lobes followed by straining through a 30 um nylon gauze (Sefar Ltd., UK) or a 40 um cell strainer. To obtain single-cell suspensions of lymphocytes from adult mice, peripheral lymph nodes (axillary and inguinal), thymus and spleen were dissected and strained through a 100 um cell strainer. Livers, lungs and kidneys were dissected and cut into pieces. Small intestines were dissected, flushed with ice-cold PBS, cut open longitudinally and into pieces. Organ pieces were digested in RPMI supplemented with 10% FBS containing 1 mg/ml collagenase type IV (Roche) and 100 ug/ml DNase I (Sigma) for 30 min shaking at 37°C, followed by filtering through an 100 um cell strainer. Cells were resuspended in a 40% isotonic Percoll solution and centrifuged on a 80% Percoll solution for 20 min at 700 x g at room temperature with brake off. Leukocytes were recovered from the interface, resuspended, and used for further analyses. Erythrocytes from blood, spleen, liver, lung and kidney samples were osmotically lysed in ACK lysis buffer (Invitrogen), and cells were washed in FACS buffer. Rosa26-hIL-7R.huCD2-Cre leukaemic thymus and lymph node cells were isolated as described, homogenized in HI-FBS containing 10% dimethylsulfoxide (DMSO) and frozen at -80 °C until used. Samples were then thawed, washed and homogenized in complete RPMI-1640. |
| Instrument | Sorts were performed on FACS AriaII and FACS AriaIII cell sorters (BD). Samples were acquired using an LSR-II and LSRFortessa X-20 flow cytometer (BD) or Canto II (BD) |
| Software | Analysis of Flow Cytometry data was performed using FlowJo. |
| Cell population abundance | Post-sort purity of gd T cell populations from pooled adult LN and spleen of WT or TCRa-/- mice was determined by re-run and recording of aliquots of purified populations on the flow cytometer. Analysis of re-run data acquired confirmed purities of from >90% to >99%. |
| Gating strategy | The gating strategy to determine abundance of CD8-expressing gd T cell subsets within organs and FTOC was the following: 1. SSC-A/time, 2. lymphocytes gate by FSC-A/SSC-A, 3. single cell gating and doublet discrimination (FSC-A/FSC-H and SSC-A/SSC-H), 4. live vs dead cells, 5. gating on gd T cells (either CD3+/TCRd+ or TCRd+/TCRb- or TCRd+/SSC), 6. CD8a/CD8b or Ly6a/CD8b. The gating strategy is displayed as Extended Data Fig. 1A. |

☒ Tick this box to confirm that a figure exemplifying the gating strategy is provided in the Supplementary Information.

