## [Peer Review File · Nature Immunology]

Peer Review Information

Journal: Nature Immunology

Manuscript Title: Perinatal thymic-derived CD8 $\alpha\beta$ -expressing $\gamma\delta$ T cells are innate interferon- γ producers that expand in IL-7R/STAT5B-driven neoplasms

Corresponding author name(s): Professor Daniel Pennington, Dr Gina Fiala

Reviewer Comments & Decisions:

Decision Letter, initial version:
--

27th Apr 2023

Dear Dr. Pennington,

We have now finished reviewing your manuscript entitled "Perinatal thymic-derived CD8 $\alpha\beta$ -expressing $\gamma\delta$ T cells are innate interferon- γ producers that predominate in IL-7R/STAT5B-driven $\gamma\delta$ T cell neoplasms", reference number NI-A35558.

Although the editors thought that the manuscript was interesting enough to send out for in-depth review, the reviewers were not in favor of publishing the paper in Nature Immunology because of the strength of the novel conclusions that can be drawn at this stage. Although they expressed some interest and think there are good data here, both reviewers also noted that there are two somewhat disparate parts to the paper, which is a problem for us in terms of imagining what a revised paper would look like. As such, we cannot offer a standard revision here as we cannot instruct you clearly how to solve the issues raised by the reviewers. However, we would be open to an appeal if that is something you wish to pursue and think you can address all concerns fully, something that might require a total overhaul of the paper.

Please also note - in case reviewer 2 overall feelings about the paper are not clear from comments to the authors - they did elaborate more in the comments to editors. To paraphrase reviewer 2, they pointed out to us that the paper really needs better evidence for CD8ab+ gamma delta T cell function in infection and/or tumor rejection and each part is presently preliminary.

We realize that this is disappointing. I hope that you continue to consider Nature Immunology for your results most significant for the immunology community and wish you well in your future investigations.

Sincerely,

Nick Bernard, PhD
Senior Editor
Nature Immunology

Reviewers' comments:

Reviewer #1 (Remarks to the Author):

Sumaria, Fiala, et al. present an exciting ms describing the observation in two complementary PI3K-tg mouse models, based on Sumaria et al. *Sci Signal* 2021 (ref 15), reduced PI3K signaling during early prenatal gd T cell development promotes CD8ab+ gd T cells with non-terminally differentiated phenotype.

The second important observation is that pre-leukemic or malignant T cells that undergo supraphysiologic T cell proliferation in previously described IL7R overexpressing or STATB_N642H T mice are mainly CD8+ gd T cells.

As the ms stands, it is less clear how far these two observations are linked and whether this is relevant for human gd T cell-derived neoplasms, as claimed by the authors.

Thus, a few parts of the ms should be improved before publication in NI:

1. On page 5, the authors suggest to analyze the developmental trajectory. Wouldn't this be also a task for further analyzing the scRNA-seq data, e.g., by RNA-velocity or similar tools?
2. Figure 2D: Stats shown in right panels are unclear. It seems like addition of IL-4 did not significantly alter the number of CD8b+ cells nor Eomes+ cell frequencies. What would be the result if the two independent experiments are combined?
3. Fig 3 and minute amounts of CD8a/CD8b DP gd T cells in general: The authors are truly experts in their field and should be able to unambiguously identify gd T cells. However, it would be comforting to show this in the supplementary figures or to at least mention it in the methods section. Where gd T cells identified via gating CD3e+TCRgd(GL3)+ cells? Maybe some controls relying on more stringent gating on CD3e+ TCRgd(GL3)+TCRb(H57)negative cells would further support their case. This is a critical point as any very contamination from ab T cells would have a huge impact on the data. 10% of CD8ab+ gd T cells in adult WT LN seem exaggerated. Nevertheless, data from Tcra-deficient mice and the malignant or pre-clinically expanded CD8+ gd T cells described in Fig 4 and Fig5 clearly demonstrate the potential of gd T cells to express CD8ab. Please explain or discuss.
4. This manuscript should be discussed in relation to a recent study on human CD8ab+ gd T cells, namely <https://doi.org/10.1126/sciimmunol.ade3525> which found an increased frequency of circulating CD8+ $\gamma\delta$ T cells associates with chronic inflammatory conditions of diverse etiologies. Do these mouse CD8+ gd T cells also show attenuated TCR mediated responses but express CD16 and have a robust ADCC response, similar to those in humans?
5. Do the leukemia blasts in Fig 4/ Fig5 eagerly produce IFN-g to support their origin from fetal or

perinatal CD8+ gd T cells?

6. Would be convincing to show a direct link to human gd T cell malignancy. Are the malignant gd T cells cells in HSTCL also often CD8ab+?

Minor

- Fig 1E: Any difference in TCR γ usage between WT and p110d KO FTOC? What antibody was used to identify Vg2/3+ cells? No stats are given, only one representative of two experiments, only WT.
- Fig 2A: Do the authors suggest that there is some CD8b expression in CD8aa subset? Please clarify
- Axis labeling in Fig 5 is often unclear without reading the legends, e.g., what is meant by "% of gd + cells" ?
- Page 4: One "in" is unnecessary in "... were significantly expanded in number in 8-day ..."
- Anti-TCRb mAbs are used in Fig 4C but not mentioned in the methods
- Just curious, what is the function of CD8ab dimers in gd T cells that are not MHC-restricted?

Reviewer #2 (Remarks to the Author):

The authors revisit a poorly characterized population gd T cells that produce IFN γ and express CD8ab heterodimers for more detailed examination, in particular the issue of their intrathymic developmental requirements. These cells are present in the fetal/neonatal thymus and persist into adulthood, populating multiple tissues in newborn mice. The optimal development of CD8b(+) gd T cells is regulated by relatively low TCR signal strength and IL-4 and IL-7. Excessive signaling through the IL-7R/STAT5B pathway leads to an increase in these cells, which can dominate in certain mouse T cell leukemia models. The work is succinctly and clearly described and consists of two major themes: developmental processes and relevance in leukemias arising from altered IL-7 and/or STAT5 activity. There is a mismatch in expected end goals in each part, rendering the connection between the parts tenuous. In the first there are no studies designed to determine specific function of the subset, and in the second it is difficult to appreciate what physiological developmental setups make the cells susceptible to transformation and how close the situation in mouse tumor models mimic human T-ALL.

Major concerns

- It is now well established that the neonatal immune system is biased to generate gd and ab T cells that mediate rapid, innate-like, effector function, especially in tissues. This study clearly establishes that CD8ab+ gd T cells is another subset that make up the neonatal innate-like T cell compartment, perhaps paralleling neonatal innate-like CD8+ ab T cells (Smith et al. Cell 2018). Fetal CD8ab+ gd T cells were described in 1992 (Leclercq et al. Eur Immunol), which also reported the ability of IL-7 to expand them. This work needs to be cited. It would be important to place the functional relevance of these cells in the context of other IFN γ producing innate-like T cells.
- The strength of the work is the careful dissection of TCR and cytokine requirements for CD8ab+ gd T cells. However, most of the developmental studies use cells from FTOCs, which is a limitation without extensive validation of ex vivo counterparts. Data from transcriptome analyses of CD8b+ gd T cells

from FTOCs are used to infer developmental progression and potential function. The results require confirmations in ex vivo counterparts (at minimum the central markers discussed in the paper). Absence of parallel data of ex vivo counterparts for signature markers of the subset (e.g. Eomes) is perplexing.

- The functional relevance of CD8ab+ gd T cells in a physiological setting is unexplored, as the study design does not involve in vivo studies (other than tumor models), which constrains impact.
- CD8ab expression is purely considered as a marker of cell subset and there is no insight into the relevance of CD8ab coreceptor for gd subset function or thymic selection. This limitation, and other parts of study that lacks mechanistic insights, gives the impression of incompleteness, leaning towards being overtly descriptive.
- The link between excessive IL-7-mediated signaling and gd T cell leukemogenesis is interesting, but as presented, the data is largely correlative and it remains unclear how the models approximate human T cell leukemia

Other issues:

- Fig. 1D. Data of cells from untreated lobes need to be provided as control.
- Fig. 1E should show the Vg chain repertoires of all CD8b+gd T cell subsets to assess maturation dependent repertoire restriction. It would also help to have Vd chain repertoire analyses.
- For well-rounded interpretation of results of Fig.2 analysis of CD8b+ gd T cells of IL4^{-/-} or IL4R^{-/-} mice (and associated FTOC studies) would be needed.
- Do CD8b(+) gd T cells in vivo have increased phosphoSTAT5 in age-dependent manner?
- Fig. 5C needs absolute numbers.

Decision Letter, first revision:

Dear Professor Pennington,

Thank you for your letter asking us to reconsider our decision on your manuscript, "Perinatal thymic-derived CD8 α β -expressing $\gamma\delta$ T cells are innate interferon- γ producers that predominate in IL-7R/STAT5B-driven $\gamma\delta$ T cell neoplasms".

Now that I have had a chance to discuss the matter carefully with my colleagues, I am happy to say that we would consider sending your manuscript to external review if you provide the revisions outlined in your letter and they address the reviewer criticisms. I'm sure, however, that you'll understand that we cannot predict the outcome of the review process.

Once you have made these revisions, please use the URL below to submit the revised manuscript with figures and a revised version of the life sciences reporting summary. It will be available to referees (and, potentially, statisticians) to aid in their evaluation if the manuscript goes back for peer review. A revised checklist is essential for re-review of the paper.

The Reporting Summary can be found here:

The Editorial Policy Checklist can be found here: <https://www.nature.com/documents/nr-editorial-policy-checklist.pdf>

[REDACTED]

Please let us know how you wish to proceed and when we can expect your revised manuscript.

With kind regards,

Nick Bernard, PhD
Senior Editor
Nature Immunology

Author Rebuttal, first revision:

See inserted PDF

NI-A35558 (Sumaria, Fiala et al.) – Point-by-Point Reply:

We thank the editor and the reviewers for their knowledgeable comments, which have led to new experiments, substantial additional data, and a major restructuring of our paper. Here we provide a point-by-point reply to all the issues that had been raised on our initial submission.

General Comments

Editor: “...there are two somewhat disparate parts to the paper, which is a problem for us in terms of imagining what a revised paper would look like...As such... it might require a total overhaul of the paper...”

Reviewer #1: “...it is less clear how far these two observations are linked...”

Reviewer #2: “...There is a mismatch in expected end goals in each part, rendering the connection between the parts tenuous...”

Our response: We have restructured the paper by providing two new main figures (Fig. 1 and Fig. 6) plus additional data at the transition between the previous “disparate parts” – Fig. 4 – **to overcome the following limitations of our initial submission:**

- Lack of *in vivo* data on the contribution of CD8 $\alpha\beta^{+}$ $\gamma\delta$ T cells to immune responses. **Fig. 1** now demonstrates their expansion and IFN γ production upon malaria and tumour challenges, where they account for up to half of the IFN γ -producing $\gamma\delta$ TILs.
- Lack of human (patient) data to support the clinical significance of CD8 $\alpha\beta^{+}$ $\gamma\delta$ T cells. **Fig. 6** now shows mRNA and protein data that identifies a new subtype of CD8 $\alpha\beta^{+}$ $\gamma\delta$ T-ALL (T-cell acute lymphoblastic leukaemia).
- Weak connection between the thymic development and the leukaemia sections. The substantially revised **Fig. 4** now shows; side-by-side the impact of IL-4 and IL-7 on normal development of CD8 $\alpha\beta^{+}$ $\gamma\delta$ T cells; the observation that IL-7 is the main driver of CD8 $\alpha\beta^{+}$ $\gamma\delta$ T cell expansion in FTOC; and the selective high expression of IL-7R by CD8 $\alpha\beta^{+}$ $\gamma\delta$ T cells (compared with other $\gamma\delta$ T cell subsets). This constitutes a better transition from the analysis of a key factor (IL-7) in the normal development of CD8 $\alpha\beta^{+}$ $\gamma\delta$ T cells to the investigation of the IL-7/IL7R-mediated leukaemic potential of this perinatal subset in the context of paediatric disease (T-ALL).

Overall, we believe the revised manuscript flows much better and clarifies the added value of dissecting, not only the normal, but also the malignant development of this poorly understood subset of $\gamma\delta$ T cells, with translational relevance for human T-ALL.

Comments: Reviewer #1

R1Ma#1 comment: “On page 5, the authors suggest to analyze the developmental trajectory. Wouldn't this be also a task for further analyzing the scRNA-seq data, e.g., by RNA-velocity or similar tools?”

Our response: We thank the Reviewer for this suggestion, which has led to a significantly revised **Fig. 3** (see panels **F-I**). As the reviewer surmised, the use of “pseudotime” tools to

analyse our RNA-seq data was revealing. *Slingshot* analysis clearly suggests that CD8 $\alpha\beta^{(+)}$ $\gamma\delta$ T cells develop along a different trajectory to other IFN-pathway-associated $\gamma\delta$ T cell populations (e.g., precursors of V γ 5V δ 1 $^{(+)}$ DETC). Moreover, it also suggested that the terminally-differentiated thymic state of CD8 $\alpha\beta^{(+)}$ $\gamma\delta$ T cells is one in which they display a prominent type-I interferon-responsive gene signature, in addition to the innate-like gene signature they share with innate-like CD8 $\alpha\beta^{(+)}$ $\alpha\beta$ T cells.

R1Ma#2 comment: “Figure 2D: Stats shown in right panels are unclear. It seems like addition of IL-4 did not significantly alter the number of CD8b+ cells nor Eomes+ cell frequencies. What would be the result if the two independent experiments are combined?”

Our response: The Reviewer had identified an error in this figure. One of the bars indicating significance had been displaced. It should have been showing a statistically significant increase of Eomes $^{(+)}$ cells in the presence of IL-4 (note that there is no difference in Eomes $^{(+)}$ cells in the presence of U0126 as was previously indicated on this graph). The corrected version of this figure panel is shown below, now with more data points added from further experimental repeats. In the new version of the manuscript, these developmental data have been supplemented to additionally demonstrate the role of IL-7, in a new Fig. 4D & 4E.

Please see below the corrected version of former Fig. 2D. These data are now included in new Fig. 4D & 4E:

R1Ma#3 comment: “Fig 3 and minute amounts of CD8 α /CD8 β DP $\gamma\delta$ T cells in general: The authors are truly experts in their field and should be able to unambiguously identify $\gamma\delta$ T cells. However, it would be comforting to show this in the supplementary figures or to at least mention it in the methods section. Were $\gamma\delta$ T cells identified via gating CD3 $\epsilon^{(+)}$ TCR $\gamma\delta$ (GL3) $^{(+)}$ cells? Maybe some controls relying on more stringent gating on CD3 $\epsilon^{(+)}$ TCR $\gamma\delta$ (GL3) $^{(+)}$ TCR β (H57)negative cells would further support their case. This is a critical point as any contamination from $\alpha\beta$ T cells would have a huge impact on the data.”

Our response: We now include our stringent gating strategy, with exclusion of TCR β -expressing cells, as the new **Sup. Fig. 1A**. The gating strategy used to identify $\gamma\delta$ T cells (subsequently analysed for surface expression of CD8 α and CD8 β), is shown below: the blue arrows show the chronology of the gating strategy starting with a time gate, followed by gating thymocytes/lymphocytes based on forward scatter-area (FSC-A) vs side scatter-area (SSC-A), removal of multiplets (FSC-H vs FSC-A and SSC-H vs SSC-A), gating on live cells, and finally, selection of TCR δ^+ TCR β^- cells (i.e., $\gamma\delta$ T cells).

New Sup. Fig. 1A:

Moreover, we also show that CD8 $\alpha\beta^{(+)}$ $\gamma\delta$ T cells are readily detected in $\alpha\beta$ T cell-deficient mice: In TCR $\beta^{-/-}$ and TCR $\alpha^{-/-}$ mice similar or increased proportions of CD8 $\alpha\beta^{(+)}$ $\gamma\delta$ T cells are observed in spleen compared to WT mice. TCR $\beta^{-/-}$ data are presented as part of **new Fig. 2E**:

TCR $\alpha^{-/-}$ data are presented as part of **new Sup. Fig. 2D** (see next page):

In newborn thymus, TCRα^{-/-} mice show a slightly increased relative contribution but normal numbers of CD8αβ⁽⁺⁾ γδ T cells (see below).

R1Ma#4 comment: "...10% of CD8αβ⁺ γδ T cells in adult WT LN seem exaggerated. Nevertheless, data from Tcrα-deficient mice and the malignant or pre-clinically expanded CD8⁺ γδ T cells described in Fig4 and Fig5 clearly demonstrate the potential of γδ T cells to express CD8αβ. Please explain or discuss."

Our response: The data identifying and quantifying CD8αβ⁽⁺⁾ γδ T cells (as a % of total γδ T cells), in various tissues were generated by independent researchers working at different Institutes with different mouse facilities in different countries – UK and Portugal. These data tally well; please see below for comparisons (currently not shown in the manuscript, but could be added upon reviewer/editorial request).

R1Ma#5 comment: “This manuscript should be discussed in relation to a recent study on human CD8 $\alpha\beta^+$ $\gamma\delta$ T cells, namely <https://doi.org/10.1126/sciimmunol.ade3525> which found an increased frequency of circulating CD8+ $\gamma\delta$ T cells associates with chronic inflammatory conditions of diverse etiologies. Do these mouse CD8+ $\gamma\delta$ T cells also show attenuated TCR mediated responses but express CD16 and have a robust ADCC response, similar to those in humans?”

Our response: This interesting paper, which came out a week after the initial submission of our manuscript, is now cited in both the introduction and the discussion. As we had shown in our initial submission, mouse CD8 $\alpha\beta^{(+)}$ $\gamma\delta$ T cells also show poor responsiveness through their TCR, instead making robust responses when treated with cytokines – please see **Fig. 2C** in the revised manuscript. Regarding **CD16**, we have now analysed its expression, **but found little to no expression** on mouse CD8 $\alpha\beta^{(+)}$ $\gamma\delta$ T cells from 8-day WT FTOC, as shown below. CD8 $\alpha\beta^{(+)}$ $\gamma\delta$ T cells from B6 WT adult lymph nodes were also negative for CD16 (not shown). By contrast, a substantial proportion of CD8 $\alpha\alpha^{(+)}$ $\gamma\delta$ T cells and CD8 $^{(-)}$ $\gamma\delta$ T cells in the foetal thymus expressed CD16. These data are currently not shown in the manuscript but could be added upon reviewer/editorial request.

R1Ma#6 comment: “Do the leukaemia blasts in Fig 4/Fig5 eagerly produce IFN- γ to support their origin from fetal or perinatal CD8+ $\gamma\delta$ T cells?”

Our response: Indeed, we found that ~95% of *Vav1.hSTAT5B^{N642H}* transgenic CD8 $\alpha\beta^{(+)}$ $\gamma\delta$ T cells produce IFN γ (compared to ~40% of “WT” controls), as shown in the **new Sup. Fig. 6G**, suggesting the targeted dysregulation of this perinatal IFN γ -biased $\gamma\delta$ T cell subset.

R1Ma#7 comment: “would be convincing to show a direct link to human $\gamma\delta$ T cell malignancy” (...)
“it is less clear whether this is relevant for human $\gamma\delta$ T cell-derived neoplasms”

Our response: We thank the Reviewer for this important comment. We have established two new collaborations with clinical centres in order to obtain human/patient $\gamma\delta$ T-ALL data. The collaboration with Charles Mulligan (St. Jude’s, Memphis, USA), allowed us to examine their RNA-seq data from a collection of T-ALL cases and find 3 out of 20 $\gamma\delta$ T-ALL cases positive for both CD8B and CD8A; and, interestingly, these expressed high levels of IL7R (**new Fig. 6A**).

To complement this, the collaboration with Ludovic Lhermitte (Necker, Paris, France) provided cell surface protein expression, i.e. flow cytometry data, specifically for CD8 β (since until now, only CD8 α expression has been routinely assessed in hematology laboratories). In their independent cohort, another two CD8 $\beta^{(+)}$ $\gamma\delta$ T-ALL cases were identified (**new Fig. 6C**). Collectively, these data estimate a ~5-15% prevalence of CD8 $\beta^{(+)}$ $\gamma\delta$ T-ALL, thus defining it as a **new subtype of human neoplasm**.

Minor points

R1Mi#1 comment: “Fig 1E: Any difference in TCR γ usage between WT and p110d KO FTOC?”

Our response: We found no significant differences in TCR-V γ usage by CD8 $\alpha\beta^{(+)}$ $\gamma\delta$ T cells between WT and p110 $\delta^{-/-}$ FTOC, as shown below. The far left pie-charts show *total* CD8 $\alpha\beta^{(+)}$ $\gamma\delta$ T cells whereas the next 3 sets of pie-charts show CD8 $\alpha\beta^{(+)}$ $\gamma\delta$ T cells within the indicated $\gamma\delta$ developmental subset ($\gamma\delta^{24+}$, $\gamma\delta^{TN}$, and $\gamma\delta^{RB}$). These data have been added to the paper as **new Fig. 3E**.

R1Mi#2 comment: “What antibody was used to identify V γ 2/3+ cells?”

Our response: As the Reviewer is likely hinting at, there are no antibodies available that specifically detect V γ 2/3. Thus, the identification of these cells was based on a process of elimination. We have now changed the label in these pie-charts from ‘V γ 2/3’ to ‘other’ (see also Mi#1 above).

R1Mi#3 comment: “No stats are given, only one representative of two experiments, only WT.”

Our response: This comment refers to the V γ -usage pie-charts. Please see below for a comprehensive analysis (using available Abs) of V γ (A) and V δ (B) usage by CD8 $\alpha\beta^{(+)}$ $\gamma\delta$

T cells at different stages of development. These data now include the mean \pm s.d values. A selection of these data is shown as **new Sup. Fig. 3C**.

R1Mi#4 comment: “Fig 2A: Do the authors suggest that there is some CD8b expression in CD8aa subset? Please clarify.”

Our response: The CD8 $\alpha\alpha$ and CD8 $\alpha\beta$ boundaries were set using standard FACS-gating (for example, as shown in Fig 2A – and several other places). From this, it is clear that a subset of $\gamma\delta$ T cells express *only* CD8 $\alpha\alpha$ homodimers while our cells of interest express CD8 $\alpha\beta$ heterodimers.

R1Mi#5 comment: “Axis labelling in Fig 5 is often unclear without reading the legends, e.g., what is meant by “% of $\gamma\delta$ + cells”?”

Our response: Thank you for pointing this out. We have improved this labelling in the revised figure.

R1Mi#6 comment: “Page 4: One “in” is unnecessary in “... were significantly expanded in number in 8-day ...””

Our response: This has been amended in the revised manuscript.

R1Mi#7 comment: “Anti-TCR β mAbs are used in Fig 4C but not mentioned in the methods”

Our response: This information (**H57-597, APC-eFluor780, eBioscience, #47-5961-80**) has been added to the methods in the revised manuscript.

R1Mi#8 comment: “Just curious, what is the function of CD8 $\alpha\beta$ dimers in $\gamma\delta$ T cells that are not MHC-restricted?”

Our response: This is an interesting question that we have considered at length. Since CD8 $\alpha\beta$ heterodimers are best known for their capacity to bind MHC class I complexes, thus being required for the selection (and antigen recognition) of CD8 $^+$ $\alpha\beta$ T cells, we investigated if MHC class I expression was necessary for generation of CD8 $\alpha\beta^{(+)}$ $\gamma\delta$ T cells. Importantly, we found normal numbers of CD8 $\alpha\beta^{(+)}$ $\gamma\delta$ T cells in mice lacking the essential MHC class I component, $\beta 2m$, both in newborn thymus (data are currently not shown in the manuscript but could be added upon reviewer/editorial request) and adult spleen (data are shown below and as part of the **new Fig. 2E**) which suggests that CD8 $\alpha\beta$ expression on $\gamma\delta$ T cells is not required for functional interactions with MHC class I during development.

The data shown below are now part of **new Fig. 2E**:

We also observed normal IFN γ production by LN CD8 $\alpha\beta^{(+)}$ $\gamma\delta$ T cells of β 2m KO mice indicating normal functional development of these cells in the absence of MHC class I expression. This data is shown below and as part of **new Sup. Fig. 2E**.

As an alternative hypothesis, we are considering that CD8 $\alpha\beta$ might modify the TCR-mediated activation threshold of CD8 $\alpha\beta^{(+)}$ $\gamma\delta$ T cells⁽⁺⁾, since the heterodimer is known to interact avidly with the key signal transducer Lck (e.g. **PMID 14500983**). However, this will likely be rather complex to dissect, and we would kindly defer it to a follow-up study.

Reviewer #2

R2Ma#1 comment: "...there are no studies designed to determine specific function of the subset...the paper really needs better evidence for CD8 $\alpha\beta^{+}$ gamma delta T cell function in infection and/or tumour rejection."

Our response: As suggested by the Reviewer, we have performed experiments in infection and tumour models to assess the function of CD8 $\alpha\beta^{(+)}$ $\gamma\delta$ T cells. As a result, we provide a new **Fig. 1** on the contribution of CD8 $\alpha\beta^{(+)}$ $\gamma\delta$ T cells to immune responses. This demonstrates the expansion and IFN γ production of CD8 $\alpha\beta^{(+)}$ $\gamma\delta$ T cells upon malaria and tumour challenges, where (in the latter) they account for up to half of IFN γ -producing $\gamma\delta$ TILs.

We also provide new human patient data to support the clinical significance of CD8 $\alpha\beta^{(+)}$ $\gamma\delta$ T cells. **Fig. 6** now shows mRNA and protein data that identifies a new subtype of CD8 $\alpha\beta^{(+)}$ $\gamma\delta$ T-ALL (T-cell acute lymphoblastic leukaemia).

R2Ma#2 comment: “Fetal $CD8\alpha\beta^+$ $\gamma\delta$ T cells were described in 1992 (Leclerq et al. *Eur Immunol*), which also reported the ability of IL-7 to expand them. This work needs to be cited. It would be important to place the functional relevance of these cells in the context of other IFN γ producing innate-like T cells.”

Our response: We have now cited (REF#17) this paper in the introduction (page 3) of the revised manuscript. For the functional contribution of $CD8\alpha\beta^{(+)}$ $\gamma\delta$ T cells among IFN γ producers in malaria infection and breast tumour models, please see our response to the previous comment and additional data in our new Fig. 1.

R2Ma#3 comment: “The strength of the work is the careful dissection of TCR and cytokine requirements for $CD8\alpha\beta^+$ $\gamma\delta$ T cells. However, most of the developmental studies use cells from FTOCs, which is a limitation without extensive validation of ex vivo counterparts. Data from transcriptome analyses of $CD8\beta^+$ $\gamma\delta$ T cells from FTOCs are used to infer developmental progression and potential function. The results require confirmations in ex vivo counterparts (at minimum the central markers discussed in the paper). Absence of parallel data of ex vivo counterparts for signature markers of the subset (e.g., Eomes) is perplexing.”

Our response: We acknowledge these points. We have now addressed this by including analysis of $CD8\alpha\beta^{(+)}$ $\gamma\delta$ T cells ex vivo from thymus and spleen taken at various time-points during mouse ontogeny; at birth, day-4, day-8, day-10, and adult – please see below and new Sup. Fig. 4. These data show a sizable population of $CD8\alpha\beta^{(+)}$ $\gamma\delta$ T cells at all time-points, and expression of signature markers including CD8 β , Ly6a, and Eomes.

R2Ma#4 comment: “The functional relevance of CD8 $\alpha\beta$ ⁺ $\gamma\delta$ T cells in a physiological setting is unexplored, as the study design does not involve *in vivo* studies (other than tumour models), which constrains impact.”

Our response: To assess the functional contribution of mouse CD8 $\alpha\beta$ ⁽⁺⁾ $\gamma\delta$ T cells to immune responses, we employed *Plasmodium berghei* infection, a model of severe malaria driven by IFN γ -producing $\gamma\delta$ T cells (Ribot *et al.*, *PNAS* 2019, 116(20):9979-9988); and the transplantable E0771 breast tumour model (in collaboration with Seth Coffelt, U. Glasgow, UK), where we previously reported an anti-tumour role for IFN γ -producing $\gamma\delta$ T cells (Lopes *et al. Nat Immunol* 2021, 22(2):179-192). These data are now found in a revised **Fig. 1**. In brief, CD8 $\alpha\beta$ ⁽⁺⁾ $\gamma\delta$ T cells expanded upon *Plasmodium* infection particularly in spleen and liver (**Fig. 1B**). Functionally, as expected, CD8 $\alpha\beta$ ⁽⁺⁾ $\gamma\delta$ T cells produced IFN γ (rather than IL-17), for which they were particularly enriched among $\gamma\delta$ T cell subsets (**Fig. 1C**). In the orthotopic E0771 breast cancer model, CD8 $\alpha\beta$ ⁽⁺⁾ $\gamma\delta$ T cells were clearly found as tumour-infiltrating lymphocytes (TILs) (**Fig. 1E**), and importantly accounted for almost half the IFN γ -producing $\gamma\delta$ TILs, to a similar level as CD8⁽⁻⁾ $\gamma\delta$ cells, and in stark contrast with CD8 $\alpha\alpha$ ⁽⁺⁾ $\gamma\delta$ cells (**Fig. 1F**). This major contribution was selective to the tumour bed and was not found in distal lymph nodes (**Sup. Fig. 1C**), suggesting a preferential expansion of CD8 $\alpha\beta$ ⁽⁺⁾ $\gamma\delta$ T cells within E0771 tumours. Although we currently lack the tools to selectively deplete this subset *in vivo*, our data clearly demonstrate that mouse CD8 $\alpha\beta$ ⁽⁺⁾ $\gamma\delta$ T cells are strong responders to both infectious and cancer challenges, thus setting the stage for the dissection of the normal and malignant development of these cells in the remainder of the revised manuscript.

In humans, functional observations with respect to the corresponding CD8 $\alpha\beta$ ⁽⁺⁾ $\gamma\delta$ T cell subset are hampered by the limited use of anti-TCR $\gamma\delta$ together with anti-CD8 β in standard phenotyping analyses of patient material combined with the practice of gating out CD4⁺ and CD8⁺ lymphocytes on the way of identifying $\gamma\delta$ T cells. Therefore, up to now, functional observations of a clearly identified CD8 $\alpha\beta$ ⁽⁺⁾ $\gamma\delta$ T cell subset are limited to two studies. One (published after our initial submission), suggested that CD8 $\alpha\beta$ ⁽⁺⁾ $\gamma\delta$ T cells are a hallmark of chronic (but not acute) *Mycobacterium tuberculosis* infection (Roy Chowdhury *et al.*, *Sci Immunol* 2023, 8(81):eade3525). The other reports their negative correlation with inflammatory bowel disease activity (Kadivar *et al.*, *J Immunol* 2016, 197(12):4584-4592). Both of these studies are now cited in the revised manuscript, supporting our view that CD8 $\alpha\beta$ ⁽⁺⁾ $\gamma\delta$ T cells are an important T cell population in both health and disease.

R2Ma#5 comment: “CD8 $\alpha\beta$ expression is purely considered as a marker of cell subset and there is no insight into the relevance of CD8 $\alpha\beta$ coreceptor for *gd* subset function or thymic selection. This limitation, and other parts of study that lacks mechanistic insights, gives the impression of incompleteness, leaning towards being overtly descriptive.”

Our response: We have considered the role of CD8 $\alpha\beta$ at length. Since CD8 $\alpha\beta$ heterodimers are best known for their capacity to bind MHC class I complexes, thus being required for the selection (and antigen recognition) of CD8⁺ $\alpha\beta$ T cells, we investigated if MHC class I expression was necessary for generation of CD8 $\alpha\beta$ ⁽⁺⁾ $\gamma\delta$ T cells. Importantly, we found normal numbers of CD8 $\alpha\beta$ ⁽⁺⁾ $\gamma\delta$ T cells in mice lacking the essential MHC class I component, $\beta 2m$, both in newborn thymus (data are currently not shown in the manuscript

but could be added upon reviewer/editorial request) and adult spleen (data are shown below and as part of the **new Fig. 2E**) which suggests that CD8 $\alpha\beta$ expression on $\gamma\delta$ T cells is not required for functional interactions with MHC class I during development.

The data below are now shown as part of **new Fig. 2E**:

We also observed normal IFN γ production by LN CD8 $\alpha\beta^{(+)}$ $\gamma\delta$ T cells of $\beta 2m$ KO mice indicating normal functional development of these cells in the absence of MHC class I expression. These data are shown below and are now part of **new Sup. Fig. 2E**.

As an alternative hypothesis, we are considering that CD8 $\alpha\beta$ might modify the TCR-mediated activation threshold of CD8 $\alpha\beta^{(+)}$ $\gamma\delta$ T cells, since the heterodimer is known to interact avidly with the key signal transducer Lck (e.g. **PMID 14500983**). However, this will likely be rather complex to dissect, so we would kindly defer it to a follow-up study.

R2Ma#6 comment: “The link between excessive IL-7-mediated signalling and $\gamma\delta$ T cell leukemogenesis is interesting, but as presented, the data is largely correlative, and it remains unclear how the models approximate human T cell leukaemia”, and “...it is difficult to appreciate what physiological developmental setups make the cells susceptible to transformation and how close the situation in mouse tumour models mimic human T-ALL...”

Our response: We thank the Reviewer for this important comment. We have established two new collaborations with clinical centres in order to obtain human/patient $\gamma\delta$ T-ALL data. The collaboration with Charles Mulligan (St. Jude’s, Memphis, USA), allowed us to examine their RNA-seq data from a collection of T-ALL cases and find 3 out of 20 $\gamma\delta$ T-ALL cases positive for both CD8B and CD8A; and, interestingly, these expressed high levels of IL7R (new Fig. 6A).

To complement this, the collaboration with Ludovic Lhermitte (Necker, Paris, France) provided cell surface protein expression, i.e. flow cytometry data, specifically for CD8 β (since until now, only CD8 α expression has been routinely assessed in hematology laboratories). In their independent cohort, another two CD8 $\beta^{(+)}$ $\gamma\delta$ T-ALL cases were identified (new Fig. 6C). Collectively, these data estimate a ~5-15% prevalence of CD8 $\beta^{(+)}$ $\gamma\delta$ T-ALL, thus defining it as a **new subtype of human neoplasm**.

Other issues:

R2Mi#1 comment: “Fig. 1D. Data of cells from untreated lobes need to be provided as control.”

Our response: We are slightly confused by this comment as thymic lobe data from Fig. 1D (now Fig. 3A in the revised manuscript) was from WT mice *ex vivo* at the ages indicated.

R2Mi#2 comment: “Fig. 1E should show the $V\gamma$ chain repertoires of all CD8 β^{+} $\gamma\delta$ T cell subsets to assess maturation dependent repertoire restriction. It would also help to have $V\delta$ chain repertoire analyses.”

Our response: We thank the Reviewer for this suggestion. We now provide a comprehensive analysis of $V\gamma$ -usage, and a partial analysis of $V\delta$ -usage (with available antibodies for $V\delta 4$ and $V\delta 6$), by CD8 $\alpha\beta^{(+)}$ $\gamma\delta$ T cells at different stages of development ($\gamma\delta^{24+}$, $\gamma\delta^{TN}$, $\gamma\delta^{RB}$ subsets etc) in FTQC. Although “waves” of $V\gamma$ -usage are observable, there is no obvious restriction of particular $V\gamma$ -chains, suggesting that TCRs incorporating all $V\gamma$ -chains

(and likely $V\delta$ - although antibodies are limiting), are permissible for development of $CD8\alpha\beta^{+} \gamma\delta$ T cells. A selection of these data is shown below and in new Sup. Fig. 3C.

R2Mi#3 comment: “For well-rounded interpretation of results of Fig.2 analysis of CD8 β ⁺ $\gamma\delta$ T cells of IL4^{-/-} or IL4R^{-/-} mice (and associated FTOC studies) would be needed.”

Our response: We thank the Reviewer for raising this point. We have obtained IL-4R α ^{-/-} mice from Prof. Judi Allen (Uni. Manchester) and observed a reduced CD8 $\alpha\beta$ ⁽⁺⁾ $\gamma\delta$ T cell compartment and Eomes⁽⁺⁾ cells (in LN), as shown below and in **new Sup. Fig. 5B**.

Moreover, we performed blocking experiments with an anti-IL-4 antibody in FTOC. As shown below, anti-IL-4 antibody alone did not significantly alter the number of CD8 $\alpha\beta$ ⁽⁺⁾ $\gamma\delta$ T cells or Eomes expression, but it did so in “ideal” conditions for development of CD8 $\alpha\beta$ ⁽⁺⁾ $\gamma\delta$ T cells; i.e., with additional IL-4 plus UO126 inhibitor. These data are currently not shown in the manuscript but could be added upon reviewer/editorial request.

R2Mi#4 comment: Do CD8 β (+) $\gamma\delta$ T cells *in vivo* have increased phosphoSTAT5 in age-dependent manner?

Our response: With regard to phosphoSTAT5, we observed higher levels at 16 months for all $\gamma\delta$ T cell subsets upon stimulation with IL-7 (50 ng/ml, 1h) – see below. Although CD8 $\alpha\beta$ (+) $\gamma\delta$ T cells showed higher levels of pSTAT5 than the other subsets at all ages, the differences were small, and therefore we have not added these data to the revised manuscript:

R2Mi#5 comment: Fig. 5C needs absolute numbers.

Our response: The corresponding absolute numbers were depicted in former **Supplementary Fig 4B**, which are now included in **Fig. 5C** in the revised manuscript.

Decision Letter, second revision:

25th Mar 2024

Dear Dr. Pennington,

Thank you for submitting your revised manuscript "Perinatal thymic-derived CD8 $\alpha\beta$ -expressing $\gamma\delta$ T cells are innate interferon- γ producers that expand in immune responses and in IL-7R/STAT5B-driven neoplasms" (NI-A35558B). It has now been seen by the original referees and their comments are below. The reviewers find that the paper has improved in revision, and therefore we'll be happy in principle to publish it in Nature Immunology, pending minor revisions to satisfy the referees' final requests and to comply with our editorial and formatting guidelines.

We will now perform detailed checks on your paper and will send you a checklist detailing our editorial and formatting requirements in about a week. Please do not upload the final materials and make any revisions until you receive this additional information from us.

If you had not uploaded a Word file for the current version of the manuscript, we will need one before beginning the editing process; please email that to immunology@us.nature.com at your earliest convenience.

Thank you again for your interest in Nature Immunology Please do not hesitate to contact me if you have any questions.

Sincerely,

Nick Bernard, PhD
Senior Editor
Nature Immunology

Reviewer #1 (Remarks to the Author):

The authors Sumaria, Fiala, et al. have substantially revised the structure and storytelling of this ms. Now the connection of normal and malignant development of CD8ab(+) gd T cells is described in a more natural flow and backed up by new clinical human/patient gd T-ALL data.

I only have some very minor points that may improve the ms

1. provide more information (non-gd17 tumour infiltrating lymphocytes (TILs)) for the gating strategy of data shown in Fig 1B and show representative plots (i.c. IFN-g+ TILs) for the eventually more delicate gating strategy of data shown in Fig 1C , maybe as part of Fig S1
2. Also, the additional analysis of ab T cell deficient mice, e.g., new Fig 2D reassures that the CD8ab+ gd T cells are not contaminated by ab T cells.

Interestingly, the authors seem to find a few CD8+ ab T cells in the Trca KO mouse? Please comment.

3. In their rebuttal, the authors state that Regarding CD16, we have now analysed its expression, but found little to no expression on mouse CD8ab(+) gd T cells from 8-day WT FTOC, as shown below. CD8ab(+) gd T cells from B6 WT adult lymph nodes were also negative for CD16 (not shown). By contrast, a substantial proportion of CD8aa(+) gd T cells and CD8(-) gd T cells in the foetal thymus expressed CD16. These data are currently not shown in the manuscript but could be added upon reviewer/editorial request.

◇ I think it would indeed be useful to add these data.

Reviewer #3 (Remarks to the Author):

The revised manuscript of Sumaria, Fiala et al manages to connect the (1) description of CD8ab+ gdT cells found in mice to their (2) physiological function and to their (3) pathophysiological development. The link to the human T-ALL data (Figure 6) remains slightly disconnected. Recent data on CMV-reactive CD8a+ gdT cells helps this connection, however the connection remains suggestive. The inclusion of two functional studies regarding their role in either tumor and malaria control, represented in Figure 1 is convincing and definitely closes the gap between description and function of these T cells. It is not convenient though that data representation is not synchronized between TILs and malaria reactive T cells. It would help the reader to show similar gating strategies and showing both percentages as well as absolute numbers of CD8ab+ INFg+ gdT cells just as shown on the TIL subset.

The newly included data showing normal number and INFg production of CD8ab+ gdT cells in b2m lacking mice is strongly suggests that the CD8ab coreceptor on gdT cells possibly functions differently than abT cell counterparts and not involved in thymic selection . As such, it provides sufficient data; however, including the thymic data in the manuscript would further strengthen this statement.

The human T-ALL data provides an interesting link between the observations in the mouse development models and human T cell leukemia. It would be however important to highlight in the text what indications the authors can point out in CD8ab+ gdT-ALL human pateints that would suggest the involvement of abnormal IL-7 signaling.

R2Mi#1 response: Agree with the authors, all the lobe data come from WT mice, untreated control is not needed.

Author Rebuttal, second revision:

Point-by-Point Reply to the Reviewers of NI-A35558B – 19th April 2024.

Reviewer #1

The authors Sumaria, Fiala, et al. have substantially revised the structure and storytelling of this m/s. Now the connection of normal and malignant development of $CD8\alpha\beta(+)$ $\gamma\delta$ T cells is described in a more natural flow and backed up by new clinical human/patient $\gamma\delta$ T-ALL data.

I only have some very minor points that may improve the m/s.

1. provide more information (non-gd17 tumour infiltrating lymphocytes (TILs)) for the gating strategy of data shown in Fig 1B and show representative plots (i.c. $IFN-\gamma+$ TILs) for the eventually more delicate gating strategy of data shown in Fig 1C , maybe as part of Fig S1.

Our response: We have now included the gating strategies leading to the data shown in Fig 1B and 1C. This is now **Extended Data Fig.1B**. The top panel shows the gating strategy leading to the $CD8\alpha/CD8\beta$ plot shown in Fig 1B., and the bottom panel shows the gating strategy to select $IFN\gamma+$ $\gamma\delta$ TILs and subsequently the $CD8\alpha/CD8\beta$ plot shown in Fig 1C.

2. Also, the additional analysis of $\alpha\beta$ T cell deficient mice, e.g., new Fig 2D reassures that the $CD8\alpha\beta+$ gd T cells are not contaminated by $\alpha\beta$ T cells. Interestingly, the authors seem to find a few $CD8+$ $\alpha\beta$ T cells in the *Trca* KO mouse? Please comment.

Our response: We thank the reviewer for sharing his/her observation. We reassessed our data and found that the “ $CD8+$ $\alpha\beta$ T cells” were the result of a gating error. In line with previous published reports, there are no $CD8+$ $\alpha\beta$ T cells in *Trca* KO mice. We adjusted our gating of $\alpha\beta$ T and $\gamma\delta$ T cells (see below) and updated current Extended Data Figure 2F accordingly.

3. In their rebuttal, the authors state that regarding CD16, we have now analysed its expression, but found little to no expression on mouse $CD8\alpha\beta^{+}$ $\gamma\delta$ T cells from 8-day WT FTOC, as shown below. $CD8\alpha\beta^{+}$ $\gamma\delta$ T cells from B6 WT adult lymph nodes were also negative for CD16 (not shown). By contrast, a substantial proportion of $CD8\alpha\alpha^{+}$ $\gamma\delta$ T cells and $CD8^{-}$ $\gamma\delta$ T cells in the foetal thymus expressed CD16. These data are currently not shown in the manuscript but could be added upon reviewer/editorial request. I think it would indeed be useful to add these data.

Our response: We have added CD16 data to **Extended Data Fig. 2** (new panel – **Extended Data Fig. 2A**).

Reviewer #2

The revised manuscript of Sumaria, Fiala et al manages to connect the (1) description of $CD8\alpha\beta^{+}$ $\gamma\delta$ T cells found in mice to their (2) physiological function and to their (3) pathophysiological development. The link to the human T-ALL data (Figure 6) remains slightly disconnected. Recent data on CMV-reactive $CD8\alpha^{+}$ $\gamma\delta$ T cells helps this connection, however the connection remains suggestive. The inclusion of two functional studies regarding their role in either tumor and malaria control, represented in Figure 1 is convincing and definitely closes the gap between description and function of these T cells. It is not convenient though that data representation is not synchronized between TILs and malaria reactive T cells. It would help the reader to show similar gating strategies and showing both percentages as well as absolute numbers of $CD8\alpha\beta^{+}$ $INF\gamma^{+}$ $\gamma\delta$ T cells just as shown on the TIL subset.

Our response: The analysis of IFN γ -producing $\gamma\delta$ T cells in malaria is now synchronized with the analysis of $\gamma\delta$ TILs. In Fig. 1F., we show CD8 α and CD8 β staining of gated IFN γ + $\gamma\delta$ T cells and analyses of percentages as well as total cell counts. The analysis of IFN γ MFI is adjusted to this gating.

The newly included data showing normal number and INF γ production of CD8 $\alpha\beta$ + $\gamma\delta$ T cells in $\beta 2m$ lacking mice is strongly suggests that the CD8ab coreceptor on $\gamma\delta$ T cells possibly functions differently than $\alpha\beta$ T cell counterparts and not involved in thymic selection. As such, it provides sufficient data; however, including the thymic data in the manuscript would further strengthen this statement.

Our response: The analysis of newborn $\beta 2m$ KO thymi is now presented in Extended Data Figure 3A (see below).

The human T-ALL data provides an interesting link between the observations in the mouse development models and human T cell leukemia. It would be however important to highlight in the text what indications the authors can point out in CD8 $\alpha\beta$ + $\gamma\delta$ T-ALL human patients that would suggest the involvement of abnormal IL-7 signaling.

Our response: We added the following sentence on page 11 to clarify the implications of the results of Fig. 6A: “This is a clear indication of the involvement of aberrant IL-7/IL-7R signaling in human CD8 $\alpha\beta$ (+) $\gamma\delta$ T-ALL, given that high levels of *IL7R* in T-ALL patients are known to associate with oncogenic IL-7R–dependent signaling activation [40]”.

R2Mi#1 response: Agree with the authors, all the lobe data come from WT mice, untreated control is not needed.

Our response: Thank you for pointing this out. We have rectified this in the manuscript.

Final Decision Letter:

Dear Dr. Pennington,

I am delighted to accept your manuscript entitled "Perinatal thymic-derived CD8 $\alpha\beta$ -expressing $\gamma\delta$ T cells are innate interferon- γ producers that expand in IL-7R/STAT5B-driven neoplasms" for publication in an upcoming issue of Nature Immunology.

Over the next few weeks, your paper will be copyedited to ensure that it conforms to Nature Immunology style. Once your paper is typeset, you will receive an email with a link to choose the appropriate publishing options for your paper and our Author Services team will be in touch regarding any additional information that may be required.

Please note that *Nature Immunology* is a Transformative Journal (TJ). Authors may publish their research with us through the traditional subscription access route or make their paper immediately open access through payment of an article-processing charge (APC). Authors will not be required to make a final decision about access to their article until it has been accepted. Find out more about Transformative Journals.

Authors may need to take specific actions to achieve compliance with funder and institutional open access mandates. If your research is supported by a funder that requires immediate open access (e.g. according to Plan S principles) then you should select the gold OA route, and we will direct you to the compliant route where possible. For authors selecting the subscription publication route, the journal's standard licensing terms will need to be accepted, including self-

archiving policies. Those licensing terms will supersede any other terms that the author or any third party may assert apply to any version of the manuscript.

Your paper will be published online soon after we receive your corrections and will appear in print in the next available issue.

Also, if you have any spectacular or outstanding figures or graphics associated with your manuscript - though not necessarily included with your submission - we'd be delighted to consider them as candidates for our cover. Simply send an electronic version (accompanied by a hard copy) to us with a possible cover caption enclosed.

If you have not already done so, we strongly recommend that you upload the step-by-step protocols used in this manuscript to the Protocol Exchange. Protocol Exchange is an open online resource that allows researchers to share their detailed experimental know-how. All uploaded protocols are made freely available, assigned DOIs for ease of citation and fully searchable through nature.com. Protocols can be linked to any publications in which they are used and will be linked to from your article. You can also establish a dedicated page to collect all your lab Protocols. By uploading your Protocols to Protocol Exchange, you are enabling researchers to more readily reproduce or adapt the methodology you use, as well as increasing the visibility of your protocols and papers. Upload your Protocols at www.nature.com/protocolexchange/. Further information can be found at www.nature.com/protocolexchange/about .

Please note that we encourage the authors to self-archive their manuscript (the accepted version before copy editing) in their institutional repository, and in their funders' archives, six months after publication. Nature Portfolio recognizes the efforts of funding bodies to increase access of the research they fund, and strongly encourages authors to participate in such efforts. For information about our

editorial policy, including license agreement and author copyright, please visit
www.nature.com/ni/about/ed_policies/index.html

Sincerely,

Nick Bernard, PhD
Senior Editor
Nature Immunology